# Targeting RNA:protein interactions with an integrative approach leads to the identification of potent YBX1 inhibitors

Krystel El Hage[1]*, Nicolas Babault[2]†, Olek Maciejak[1]†, Bénédicte Desforges[1]†, Pierrick Craveur[2], Emilie Steiner[1], Juan Carlos Rengifo-Gonzalez[1], Hélène Henrie[1], Marie-Jeanne Clement[1], Vandana Joshi[1], Ahmed Bouhss[1], Liya Wang[1], Cyril Bauvais[2], David Pastré[1]*

[1]Université Paris-Saclay, INSERM U1204, Univ Evry, Structure-Activité des Biomolécules Normales et Pathologiques (SABNP), Evry, France; [2]SYNSIGHT, Evry, France

**Abstract** RNA-protein interactions (RPIs) are promising targets for developing new molecules of therapeutic interest. Nevertheless, challenges arise from the lack of methods and feedback between computational and experimental techniques during the drug discovery process. Here, we tackle these challenges by developing a drug screening approach that integrates chemical, structural and cellular data from both advanced computational techniques and a method to score RPIs in cells for the development of small RPI inhibitors; and we demonstrate its robustness by targeting Y-box binding protein 1 (YB-1), a messenger RNA-binding protein involved in cancer progression and resistance to chemotherapy. This approach led to the identification of 22 hits validated by molecular dynamics (MD) simulations and nuclear magnetic resonance (NMR) spectroscopy of which 11 were found to significantly interfere with the binding of messenger RNA (mRNA) to YB-1 in cells. One of our leads is an FDA-approved poly(ADP-ribose) polymerase 1 (PARP-1) inhibitor. This work shows the potential of our integrative approach and paves the way for the rational development of RPI inhibitors.

***For correspondence:**
krystel.elhage@unibas.ch (KEH);
david.pastre@univ-evry.fr (DP)

†These authors contributed equally to this work

## Editor's evaluation

A novel approach is introduced to modulate and/or inhibit protein-RNA interactions, based upon integration of computational techniques with cellular assays.

## Introduction

Targeting RNA:protein interactions (RPIs) critically involved in pathological mechanisms is a promising strategy to find novel classes of drug candidates that remains largely unexploited (*Einstein et al., 2021*). RPIs in cells are highly diverse encompassing interactions with messenger RNA (mRNA; *Baltz et al., 2012*), ribosomal RNA (rRNA; *Simsek et al., 2017*), and non-coding RNA (ncRNA; *Lu et al., 2019*), which are critical to fine tune the spatiotemporal gene expression. As revealed by genomic approaches (*Van Nostrand et al., 2020*; *Castello et al., 2012*), the human genome contains more than 1000 transcripts encoding RNA-binding proteins (RBPs), thus providing a large variety of inter-actions with coding or non-coding RNAs. However, while the diversity of RNA:Protein interfaces may allow the development of RPIs inhibitory molecules (*Wu, 2020*), only scarce studies have already been undertaken and were restricted to few complexes such as LIN28/let-7 (*Roos et al., 2016*; *Wang et al.,*

2018), MUSASHI (MSI)/RNA (*Minuesa et al., 2019*) and heterogeneous nuclear ribonucleoprotein A18 (hnRNP A18)/RNA (*Solano-Gonzalez et al., 2021*).

Several challenges arise from the drug discovery process such as finding a druggable pocket in RNA-binding interfaces (*Minuesa et al., 2019*), the quality of the computational models, the strategies used in the in silico screening, and the lack of experimental feedback and validation of computationally predicted inhibitors essential to orient the rational drug design procedure toward the most relevant molecules. Besides the above-listed issues, new experimental assays must be developed to screen molecules targeting RPIs which ideally would work in a cellular context and be amenable to high content screening (HCS) (*Mattiazzi Usaj et al., 2016*; *Julio and Backus, 2021*). Indeed, to find potent inhibitors of RNA:protein interfaces, previous approaches used in vitro assays such as fluorescence polarization assay complemented by pull-down experiments with cell lysates or RNA enzyme-linked immunosorbent assay (ELISA) to test the effectiveness or selectivity of few hits (*Roos et al., 2016*; *Minuesa et al., 2019*). While in vitro approaches are important to define putative hits and lead to the validation of effective compounds, deciphering whether the selected molecules are effective in a cellular context generally relies on indirect measurements using techniques such as cellular engagement thermal shift assay (CETSA) or functional assays where the putative consequences of disrupting RPIs on cellular function bear a considerable uncertainty. Indeed, multiple functions are associated to RBPs, which renders the interpretation of the results of functional assays tricky. In addition, toxicity and off-target effects are putative biases which are always difficult to get rid of, notably when using small molecules with a Kd in the low micromolar range, which is generally the case for RPI inhibitors. To fill the gap between in vitro and functional assays, cellular approaches initially used to detect protein:protein interactions (PPIs) such as fluorescence resonance energy transfer (FRET) or proximity ligation assay (PLA) have been adapted to detect RPIs (*Jung et al., 2013*; *Camborde et al., 2017*) in cells but several technical issues have hampered their application such as the requirement of an adapter to RNA in FRET and PLA, the proximity of the donor and acceptor proteins in FRET, and the use of antibodies in PLA.

The aim of this paper is to tackle these challenges by introducing an experimental assay amenable to HCS to score RPIs in cells and a drug screening approach that integrates chemical, structural, and cellular data from both advanced computational and experimental techniques for the development of small molecules that target RPIs. As an application model we chose to target Y-Box binding protein 1 (YB-1) of the YBX1 gene. As other abundant nucleic acid binding proteins, YB-1 participates in many DNA/RNA-dependent processes such as mRNA translation, splicing, transcription, long ncRNA (lncRNA) functions, and DNA repair (*Lyabin et al., 2014*). However, YB-1 is mostly a core component of untranslated messenger ribonucleoprotein particles (mRNPs) in the cytoplasm (*Singh et al., 2015*) which, according to crosslinking immunoprecipitation coupled to sequencing (CLIP) analysis (*Wu et al., 2015*), preferentially binds coding sequences and 3'-UTRs across most transcripts with a weak specificity. Since YB-1 binds to and regulates the activation of dormant mRNAs (*Budkina et al., 2021*) which are particularly enriched in gene controlling transcription (*Roos et al., 2016*), YB-1 is possibly involved in cellular decisions; and consistently, YB-1 was recently identified as one of the few key genes that control gene expression plasticity in rats subjected to caloric restriction (*Ma et al., 2020*). Interestingly, YBX1 is also one of the genes whose gene-protein expression is the most correlated in cancers vs. normal tissues (*Kosti et al., 2016*), and YBX1 was identified among the few genes in a clustered regularly interspaced short palindromic repeats (CRISPR) screen showing the highest sensitivities with broad proteome co-expression in cancer cell lines (*Nusinow et al., 2020*, Figure S4 of this reference), pointing toward a possible role for YBX1 in cancer. The involvement of YB-1 in the progression and resistance to stress and chemotherapy (*Kang et al., 2013*; *Yang et al., 2010*; *El-Naggar et al., 2019*), notably after its translocation in the nucleus in certain cancers (*Bargou et al., 1997*), has also been documented. Together, these data make YB-1 a relevant target for cancer treatment (*Lasham et al., 2013*) and a subject of ongoing research to identify YB-1 inhibitors (*Khan et al., 2014*; *Tailor et al., 2021*). Moreover, YB-1 is one of the host proteins implicated in viral replication of human immunodeficiency virus (HIV) (*Jung et al., 2018*; *Poudyal et al., 2019*) and severe acute respiratory syndrome coronavirus 2 (SARS-CoV-2) (*Schmidt et al., 2021*) and hence targeting it along with targeting specific viral proteins can help reduce viral replication to a higher extent than just targeting the viral proteins. Our choice in targeting YB-1 was also guided by the availability of structural data on RNA:YB-1

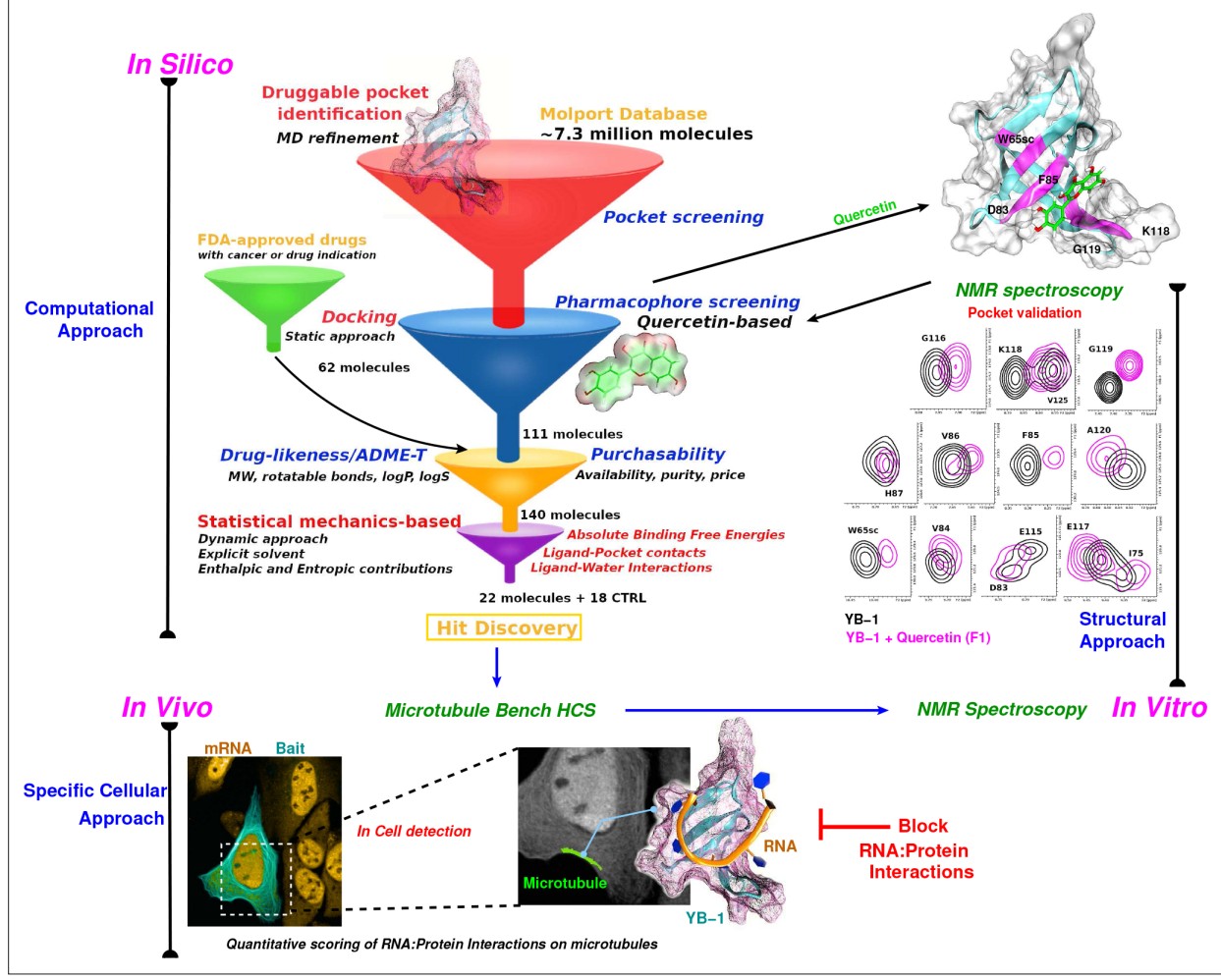

**Figure 1.** Schematic representation of an integrative approach for screening RPI inhibitors. This approach combines information from three data sources: computational (in silico, top left), cellular (in vivo, bottom), and structural (in vitro, top right). Blue arrows indicate the data flow. In silico: Starting by a large-scale computational approach that uses Docking (static approach), Molecular Dynamics and Free Energy Simulations (dynamic approach), using a computational model to virtually screen large libraries of small molecules (here, Molport and FDA-approved drugs) with the prior knowledge of a validated pocket and where several filters are used to reduce the selection to the most pertinent ligands that are then proposed as hits to be tested experimentally. Filters are represented as funnels. In vivo: In cell validation of the efficiency of the proposed hits in blocking RPIs with the MT Bench assay. This technology can quantify RPIs at the single cell level by using microtubule filaments as intracellular nanoplatforms (lower left inset, the bait, here YB-1, is shown in cyan and mRNA in orange). Lower right inset: enlarged view on mRNAs (orange) brought on microtubules using YB-1 as bait (cartoon representation: YB-1 in dark cyan with a violet surface mesh is complexed with RNA (orange ribbon)). In vitro: Experimental validation of binding the target pocket using solution NMR spectroscopy. A *zoom in* on pocket residue signals in a 2D ${}^{1}$H-${}^{15}$N-SOFAST-HMQC of YB-1 alone (black) and in the presence of Quercetin F1 (magenta). The top right 3D structure shows the binding of Quercetin (green stick) to YB-1 (cartoon representation in cyan combined with a transparent surface). Residues showing chemical shifts upon F1 binding are colored in magenta and depict what we identified as the Quercetin-pocket.

The online version of this article includes the following figure supplement(s) for figure 1:

**Figure supplement 1.** Conformational study of YB-1 in its unbound/free form using MD simulations.

**Figure supplement 2.** Structural and energetic study of YB-1:RNA (**C5**) complex using MD simulations.

complexes to probe in vitro whether small molecules can interact with the cold-shock domain (CSD) of YB-1 (*Kretov et al., 2019*; *Yang et al., 2019*).

We started this work by addressing the drug screening challenge and developing an integrative approach that uses in synergy advanced computational and experimental techniques in a concerted manner (as illustrated in *Figure 1*). Based on our discovery of a druggable pocket by molecular dynamics simulations (MD) located on the outside surface of the CSD $\beta$-barrel (which is also part of the RNA binding interface *Yang et al., 2019*), we implemented a large-scale computational approach

that balances accuracy and computational cost to virtually screen potent compounds from small molecule libraries containing more than 7 million molecules. Next, we addressed the abovementioned lack of methods able to score RPIs in a cellular context. To this end, we adapted the microtubule bench (MT bench) assay to score protein interactions with endogenous mRNAs in cells and implemented a robust HCS-based detection scheme. The MT bench was first introduced in 2015 to probe PPIs in cells with conventional fluorescence microscope by using microtubules as intracellular nanoplatforms (*Boca et al., 2015*; *Rengifo-Gonzalez et al., 2021*).

The results presented here, show that the physics-based in silico approach allowed the identification of 22 potential hits that we subsequently tested in vitro by nuclear magnetic resonance (NMR) spectroscopy and in cells using the adapted MT bench assay by scoring the interaction of YB-1 with mRNA in the cytoplasm. Of these 22 potential YB-1 inhibitors, 15 compounds were found to bind YB-1 in vitro and 11 of them were found to efficiently interfere with the interaction of YB-1 with mRNA in cells at low micromolar concentrations; and with a notable specificity when compared with two other RBPs, Human antigen R (HuR) and fused in sarcoma (FUS). The potency of the selected compounds was further demonstrated by in depth MD and NMR analyses. The results also validate that the MT bench allows to robustly and automatically score RBP-specific interactions with endogenous mRNAs by using high-resolution HCS imagers.

Interestingly, compound P1, an FDA-approved poly(ADP-ribose) polymerase 1 (PARP-1) inhibitor (*Zandarashvili et al., 2020*), was found to interact with YB-1 with higher selectivity compared to the other hits. Whether P1 interferes with YB-1 cellular functions in cells therefore merits further investigations. Together, these results demonstrate the validity of our integrative approach and the efficacy of the MT bench assay that critically complements computational and structural approaches to identify compounds targeting RPIs in cells.

## Results

### A druggable pocket found in YB-1 CSD, a conserved RNA-binding domain

The first challenge was to find a druggable pocket in the structured cold-shock domain of YB-1 located at the RNA-binding interface. We started by taking into consideration small molecules that were reported to target YB-1 in the literature. The only molecule for which a structural validation was available, though only in silico, is the flavonoid, Fisetin (*Khan et al., 2014*). In this paper, using refined docking, Fisetin was found to inhibit YB-1 activation by Akt-mediated phosphorylation at S102 with a binding pocket located inside the $\beta$-barrel structure of YB-1 CSD (51–129 aa). Having in hand the longest YB-1 fragment (1–180 aa) amenable to NMR spectroscopy (*Kretov et al., 2019*), we then analyzed the interaction between Fisetin and YB-1 fragment in vitro. Significant chemical shift perturbations (CSPs) were indeed observed but not within the previously predicted pocket (*Khan et al., 2014*). The observed CSPs implicated residues located in a hydrophobic pocket on the outside surface of the $\beta$-barrel; these are W65, V84, F85, V86, G116, K118, G119, and A120 (pocket residues shown on the top right of *Figure 1*).

Quercetin, a Fisetin analog with an additional hydroxyl group capable of forming new H-bond interactions with YB-1, was also tested. Since it showed higher CSPs with the same pocket, compared to Fisetin, we decided to subsequently name it the 'Quercetin-pocket' (average CSP of 0.032 for Quercetin (F1) compared to 0.028 for Fisetin (F4)). To delineate the characteristics of the Quercetin-pocket, we used extensive MD simulations of YB-1 CSD either in its unbound or RNA-bound form (*Figure 1—figure supplement 1* and *Figure 1—figure supplement 1*, respectively; detailed MD analysis can be found in Appendix 1). Results show that the Quercetin-pocket in its unbound form presents an open and a closed state. This pocket is located at the third β-hairpin and is monitored by K118 and F85 side chains. The opening mechanism is controlled by an electrostatic cation-π interaction formed between the cationic side chain of K118 (NH3+) and the π-electron ring system of F85 (*Figure 1—figure supplement 1*). The sampled structures of both open and closed states of CSD were also captured by NMR in the published 3D solution structure of *Kloks et al., 2002* which is consistent with our findings. MD and NMR analysis of YB-1 in complex with 5-nt long poly(C) RNA (C5) show that some of the CSD key residues implicated in RNA binding are located in the Quercetin-pocket; these residues include W65, Y72, F74, F85, H87, K118, and E121 (*Figure 1—figure supplement 1*). These

residues are evolutionary conserved as shown by the ConSurf (*Ben Chorin et al., 2020*; *Goldenberg et al., 2009*) analysis reported in Appendix 1-section III and illustrated in *Appendix 1—figure 1*. Together, MD and NMR analysis evidence the validity of the Quercetin-pocket as a potential target for the development of small molecules interfering with RNA:YB-1 interactions.

## Prediction of potent inhibitors of mRNA:YB-1 interactions using a large scale computational approach

Having identified a druggable pocket at the RNA:YB-1(CSD) interface, we next sought to target it pharmacologically. Therefore, based on these atomistic and structural data, we implemented a large-scale computational strategy to propose putative inhibitors of RNA:YB-1 interactions. This approach is illustrated in *Figure 1* and detailed in the Computational Methods section and in Appendix 2.

We started by using a pharmacophore approach to virtually screen a database composed of 208 million pharmacophores representing the conformers of around 7.3 million distinct commercially available molecules from MolPort: (i) a 'pocket-'ased" pharmacophore screening built from the prediction of a pseudo-ligand in the binding site of the MD refined structure of the open-state pocket and (ii) a distinct 'ligand-based' pharmacophore built on the 3D structure of Quercetin (F1) with YB-1. The 3D structure of the YB-1:F1 complex was obtained by docking followed by refinement MD simulations and the binding site was confirmed by NMR spectroscopy (*Figure 1*). 249 and 407 distinct molecules were selected from the 'pocket-based' and the 'ligand-based' screening, respectively. Next these molecules were reduced to a final selection by predicting ADME-T (absorption, distribution, metabolism, excretion, and toxicity) endpoints and using computed molecular docking in the Quercetin-pocket (details in Appendix 2-section I). At the end, 111 molecules were retained from this static virtual screen after visual inspection and rational selection of structurally promising candidates.

In a second step, we applied physico-chemical filters to keep only molecules belonging to a drug-like chemical space (molecular weight, number of rotational bonds, number of proton donors and acceptors, lipophilicity and solubility). Purchasability filters were also applied based on availability, purity and price in order to facilitate and optimize the conditions for the in vitro and in vivo assays. From the 7.3 million MolPort molecules, 78 molecules were finally retained. In parallel, we executed an automated blind docking of 4700 FDA-approved drugs (Drugs-lib library *Lagarde et al., 2018*) using the MTiOpenScreen web server (*Labbé et al., 2015*) which lead to the selection of 62 molecules that may target the Quercetin-pocket and may be suitable for a repositioning strategy (details in Appendix 2-sections II and III).

In the last step, the pre-selected molecules using the above static approach, 140 in total (62 FDA-approved and 78 molPort molecules), were subject to a statistical mechanics-based filter that relies on MD and free energy simulations (dynamic approach). First, the docked poses were chosen after visual inspection of the docking results (*Fischer et al., 2021*). Second, short 10 ns MD simulations were run, in the presence of explicit water molecules, in order to refine the poses and check the stability of the ligands in the targeted pocket. Only ligands that stayed in the pocket during the short MD were retained for the next step (87 out of the 140), where a weighted score ($S$) based on two observables that describes the ability of the ligand to bind and reside in the pocket was derived (this is detailed in the computational methods section). Ligands with a positive $S$ were considered as hits, and ligands with $S < 0$ were only considered as 'possible' if $S$ becomes positive when we take into account the statistical error. From the 87 molecules tested, only 26 potential hits were retained (of which 6 'possible'). Finally, absolute free energy simulations (ABFE) were used to compute the protein-ligand binding free energies ($\Delta G_{bind}$) and rank the ligands in terms of affinity (in kcal.mol– 1). ABFE simulations were done using the all-atom point charge CHARMM force field (*MacKerell et al., 1998*) and BAR (*Bennett, 1976*) for $\Delta G$ estimation. Here potential hits were selected for having a $\Delta G$ value $gt_{5.50}$ kcal/mol. However, the 6 'possible' potential hits evaluated using $S$ were considered as hits if they have a $\Delta G > 6.5$ kcal/mol (this is the case of F3: low $S$ (6.15) and high $\Delta G_{bind}$ (–10.82 kcal/mol); C11 and C12 represent a similar case). The selection of the hits at the end took into account both evaluation methods ($S$ and $\Delta G_{bind}$) and their corresponding selection criteria. For example: A3 that was not considered a hit by $S$, was considered a "possible" potential hit due to its high $\Delta G$.

Based on these criteria, 22 potential inhibitors were selected to be tested in vitro and in cells where their efficiency to inhibit mRNA:YB-1 interactions can be measured. To this list, 18 molecules, predicted inefficient, were also added as negative controls (CTRL) in order to have a total number

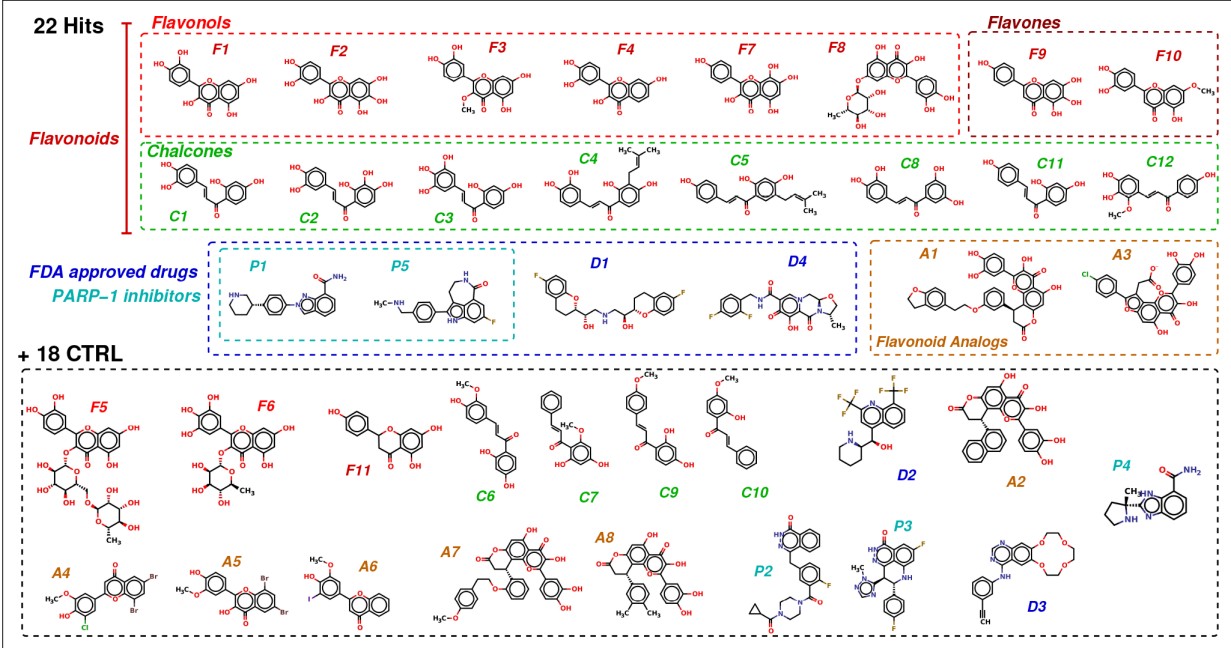

**Figure 2.** Chemical structures of the selected 40 molecules separated into 22 potential hits and 18 CTRL. Colored dashed boxes assemble hits by subclass and the black dashed box regroups the 18 CTRL. Labels and colored boxes are color coded as function of the family classification: Flavonoids (red) are divided into Flavonols (light red), Flavones (dark red) and Chalcones (green); Flavonoid Analogs in orange, FDA-approved drugs (blue) of which PARP-1 inhibitors (cyan).

The online version of this article includes the following source data for figure 2:

**Source data 1.** Classification of the 40 molecules selected using the computational approach.

of 40 molecules which is convenient for the experimental assays. However, these 18 molecules were rationally selected from the 87 molecules that stayed in the pocket and for which we have calculated and applied the statistical mechanics-based filter described above and computed their $\Delta G_{bind}$. The selection criteria was based on their structural similarity to F1 (hit validated by NMR spectroscopy) in order to generate an initial QSAR that will help us rationally optimize these molecules later. As for the FDA-approved drugs, we chose all PARP inhibitors, in order to compare with P1; the other 2 non-PARP inhibitors (D2 and D3) were chosen for their scaffold. *Figure 2* and *Figure 2—source data 1* show the classification and chemical structures of these selected 40 molecules along with their resulting scores and free energy values.

In summary, this computational approach allowed us to identify 22 potential hits from ~7 million molecule candidates.

## Robust HCS Scoring of endogenous mRPIs in cells with the MT bench assay

In order to score the interaction between mRNAs and YB-1 in cells with an HCS imager, we adapted a method that we recently developed, the MT bench (*Boca et al., 2015*). Briefly, an RBP is brought to the microtubules (MTs) after its fusion to a microtubule binding domain (MBD) so it can be used as a bait for a prey (here, mRNA). In our constructs, an RBD was fused via its C-terminus to a GFP-tag itself fused to the MBD (MBD-GFP-RBP). As MBD, we used the longest isoform of MAPT gene (2N4R-tau), which allows the binding of microtubules in a non-cooperative manner (*Butner and Kirschner, 1991*) and enables the bait protein, for example YB-1, to protrude outward the MT surface several nm away from the microtubule surface, which increases the bait accessibility to ligands *Boca et al., 2015*; the RBP brought on MTs subsequently interacts with mRNAs in the cytoplasm which results in an enrichment of endogenous mRNAs along the MT network in cells (*Figure 3a*). To measure the enrichment of poly(A)-mRNA on microtubules, we used in situ hybridization with a cy3-labeled poly(dT) probe in fixed U2OS cells (*Lubeck et al., 2014*) which have a well-extended MT network. Importantly, an HCS imager equipped with a water immersed lens (40 x, NA = 1.1) operating in confocal mode was

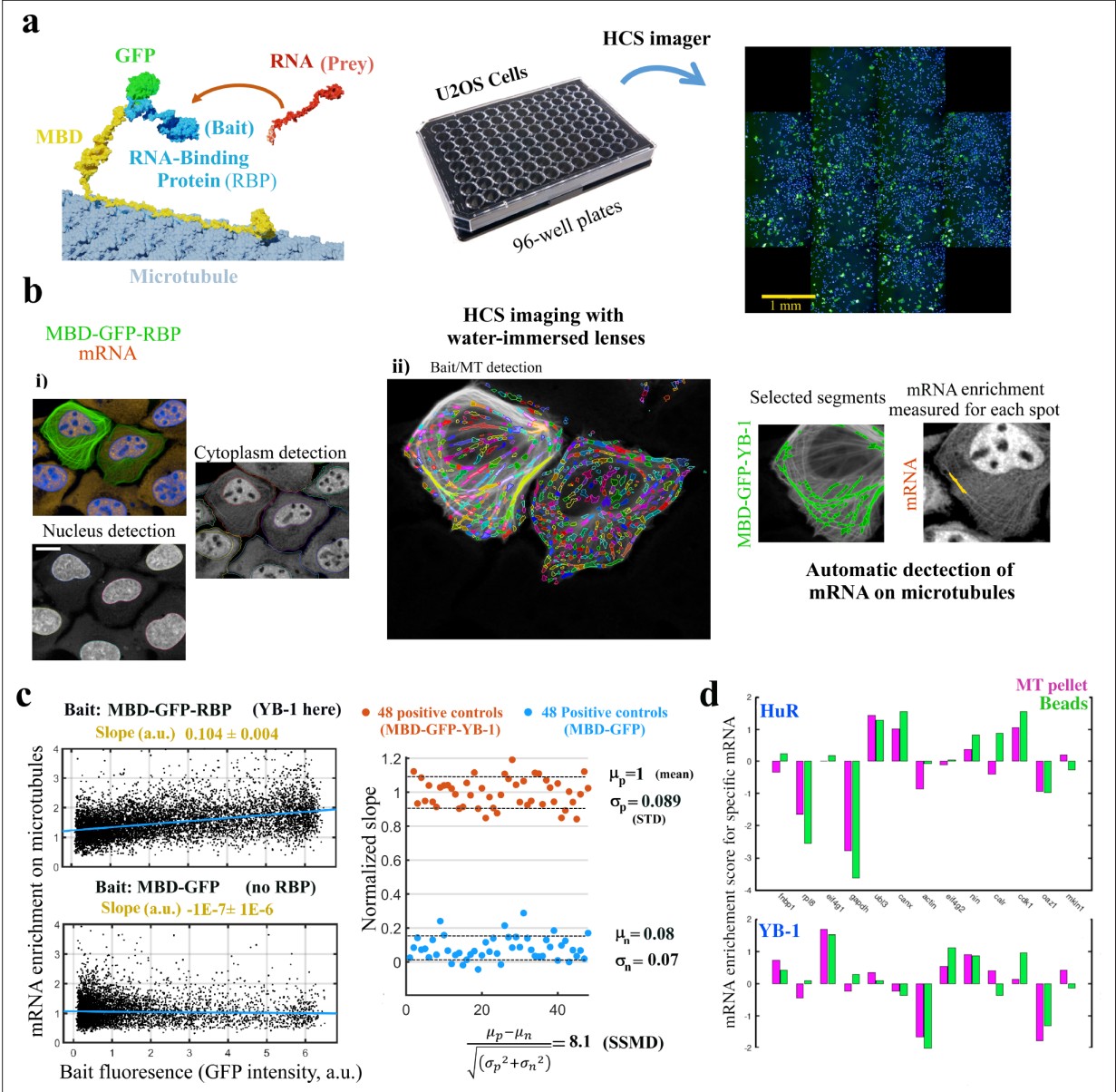

**Figure 3.** MT bench assay scores mRPIs in cells in a 96-well plate. (**a**) Left panel: Schematic view of the MT bench technology. A GFP-labeled RBP fused to MBD (Microtubule-Binding domain, yellow) was brought to microtubules in U2OS cells to attract endogenous mRNAs (in red) on the microtubule network (grey). Middle panel: Image of a 96-well plates seeded with U2OS cells. Right panel: Image of a single well processed by HCS imager showing the expression of MDP-GFP-YB-1 in U2OS cells (green). (**b**) U2OS cells expressing MBD-GFP-YB-1 (bait in green, GFP). mRNAs in red (in situ hybridization, poly(dT) probe). Nuclei in blue (DAPI). The images were obtained with an HCS imager (40 x, water immersed objective operating in confocal mode). (**i**) DAPI and the red channel (mRNA) were used to detect automatically the nuclei and cytoplasm, respectively. (**ii**) Using HARMONY "find spots" procedure, elongated spots along the microtubules were detected using the green channel (the bait, RBP). Spots were selected owing to their width-to-length ratio (<0.22) and their enrichment in GFP (YB-1). Scale bar: 20 µm. (**c**) Left panels: The enrichment of mRNAs in single selected spots (spot/cytoplasm intensity ratio, red channel) and spot bait intensity on microtubules (green channel) show a linear relationship when YB-1 was used as bait. The slope of the regression line reflects the affinity of an RBP for mRNAs. A large number of cells can be analyzed by HCS (>500 cells *per* well with in average 10–50 spots *per cell*). Slopes from linear regression were measured for each well with a 95% confidence interval. Right panel: SSMD value estimated by measuring the normalized slopes in 48 negative controls (MBD-GFP used as bait) and 48 positive controls (YB-1 was used as bait). The SSMD value is 8.1 for a 96-well plate. Spot data from all wells are shown in *Figure 3—figure supplement 2*.3a. (**d**) Bar diagram representing the enrichments of 13 different mRNAs measured by RT-PCR after two different purification procedures, co-sedimentation (MT pellet) and immunoprecipitation (Beads), and for 2 different RBPs, YB-1 and HuR; the purification procedures are illustrated in *Figure 3—figure supplement 2*.3, data and correlation analysis are provided in *Appendix 5—table 5* for 3 RBPs (YB-1, HuR, and FUS). (*continued*).

The online version of this article includes the following source data and figure supplement(s) for figure 3:

*Figure 3 continued on next page*

*Figure 3 continued*

**Figure supplement 1.** Image analysis process to quantify mRPIs in cells.

**Figure supplement 2.** Quality assessment of the MT bench cell assay.

**Figure supplement 2—source data 1.** Slope of mRNA enrichment on MTs versus bait fluorescence in selected spots for *Figure 3—figure supplement 2.3b*.

**Figure supplement 3.** DNA-binding proteins do not bring mRNAs on MTs when used as baits.

**Figure supplement 4.** The mRNAs brought onto microtubules by MBD-GFP-RBP are RBP-specific.

necessary to reach a sufficiently high lateral resolution and thus clearly distinguish the microtubule network in fluorescence microscopy images (*Figure 3b* and *Figure 3—figure supplement 2.3*). To detect the presence of baits on MTs, an automatic detection scheme has been implemented using specific criteria such as a low width-to-length ratio of the detected GFP-rich spots (<0.22) keeping only MT-shaped spots (*Figure 3b*). Details on image acquisitions and statistical analysis are provided in Appendix 3.

Results indicate an accurate detection of MBD-GFP-YB-1-decorated MTs in U2OS cells. In the selected spots, the mean bait intensity and enrichment in mRNA (ratio of the mean intensity of cy3 in the spots to that in the cytoplasm) were measured (*Figure 3c*). In contrast to MBD-GFP spots, the enrichment of mRNA in MBD-GFP-YB-1 spots located on MTs increased linearly with GFP spot fluorescence. This result demonstrates the positive correlation between the number of YB-1 brought on MTs and the relative enrichment of mRNAs on the same MTs. Interestingly, the slope thus depends directly on the binding affinity of the bait for mRNAs. We therefore considered the slope as a mRNA affinity score for RBPs brought on MTs. We next estimated the sensitivity of this scoring method by measuring the slopes of 48 positive (MBD-GFP-YB-1) and 48 negative (MBD-GFP) controls from a 96-well plate (*Figure 3c*; data from all wells are given in *Figure 3—figure supplement 2.3*). The measured SSMD value (strictly standardized mean difference) for this assay is 8.1, which is the difference of the mean values of the positive and negative controls divided by the standard deviation. A SSMD value of 8.1 corresponds to an efficient assay whatever the estimated strength of the positive controls (*Bray and Carpenter, 2017*). The SSMD value also indicates the sensitivity of the MT bench assay. Here, only molecules that decrease the slope by more than 1/8 of the positive control can be detected. Additional negative control experiments were also conducted using, as baits, 3 different DNA-binding proteins that should not bring mRNAs onto microtubules in the MT bench assay. These proteins are DNA topoisomerase 1 (TOP1), Apurinic/apyrimidinic endonuclease 1 (APE1), and DNA ligase 1 (LIG1). The results represented in *Figure 3—figure supplement 2.3* confirm that DNA-binding proteins indeed fail to bring mRNA onto the microtubules. In summary, the automatic image analysis that we implemented for the MT bench assay can reliably detect and score the interaction of YB-1 with mRNAs in the cytoplasm with HCS capacity.

## MT bench assays measure RBP-specific interactions with mRNAs in cells

Although mRNAs can be detected on microtubules in a 96-well plate setting with an HCS imager, it is critical to estimate whether fusion proteins that confine RBPs to microtubules do not lead to artificial interactions with non-specific transcripts. To this end, we designed an experiment to estimate the enrichments of mRNAs on microtubules in cells expressing MBP-GFP-RBP (mRNA brought on the microtubule with the bait protein). Briefly, cell lysates were incubated with purified MTs reconstituted in vitro from sheep brains (*Figure 3—figure supplement 2.3a*). Therefore, mRNAs were brought onto MTs owing to the presence of MBP-GFP-RBP in cell extracts and subsequently detected from MT pellets by RT-PCR after centrifugation. As a control to probe the influence of RBP confinement on microtubules, we also measured mRNA enrichments by classical RNA immunoprecipitation using magnetic beads (RIP) with anti-GFP antibody in HEK cells expressing GFP-RBP without the MBD domain. Finally, to analyze whether mRNA enrichment profiles are RBP-specific, classical RIP and microtubule co-sedimentation experiments were performed for YB-1 but also for two additional RBPs, FUS and HuR. RT-PCR analysis were performed over 13 mRNAs including actin and GADPH as abundant mRNA controls (*Figure 3d* and *Figure 3—figure supplement 2.3b and c*). Due to their high transfection efficiency, HEK cells were used to perform these experiments. Together the results show a similar profile of mRNA enrichment when the same RBP is expressed in cells, regardless of

whether classical mRNA IP or microtubule co-sedimentation was used to purify mRNAs (*Figure 3d*). In contrast, as expected since each RBP binds differentially to mRNAs, enrichment profiles are much more different when different RBPs were used as baits, regardless of the method used for mRNA purification (MT co-sedimentation or mRNA IP). Therefore, we could reasonably assume that the specific binding of RBPs to mRNAs is at least partly preserved for YB-1, FUS and HuR. However, MBD fusion and the vicinity of MTs can interfere with the binding of RBPs to certain mRNAs. For example, we do observe anti-correlations in the enrichment score for some mRNAs such as CALR mRNA (*Figure 3d*). In addition, MT co-sedimentation or mRNA IP requires cell lysis, the mRNA enrichment profile that we measured may therefore not totally reflect what is occurring in cells and in the vicinity of microtubules. Finally, the MT bench assay is obviously more adapted to detect interactions of RBPs with cytoplasmic RNAs than nuclear RNAs.

## Identification of potent mRNA:YB-1 interaction inhibitors in cells

With the additional 18 CTRL, the 22 ligands that fulfill all the above-mentioned in silico criteria were screened by using the MT bench assay. Compound concentration and level of purity were confirmed by NMR spectroscopy. These 40 molecules were then scored in two 96-well plates containing U2OS cells with four replicates *per* molecule in cells expressing MBD-GFP-YB-1. Cells were treated with 10 µM of the indicated molecules for 4 hr before fixation and analysis with the HCS imager (*Figure 4a*). Results show a significant decrease in the slope of the mRNA enrichment on microtubules versus bait expression level for 11 of the tested molecules, all of them were already considered as putative hits in silico, except C6. These 11 significant hits include 2 flavonols (F2 and F3), 7 chalcones (C1, C2, C3, C6, C8, C11, and C12), a flavonoid analog (A3) and 1 FDA-approved drug (P1) known as Niraparib, which is a PARP-1 inhibitor notably prescribed for advanced ovarian cancers. Moreover, 17 among the 18 CTRL did not lead to a significant decrease of the slope as expected. The remaining CTRL (compound C6) could be considered a false positive since it does not interact with YB-1 in vitro. To ensure that the decrease in the slope was specific to YB-1, we performed the same experiment using two other RBPs, HuR and FUS (*Figure 4b*). HuR and FUS bind to mRNA via RNA Recognition Motif (RRM), an RNA-binding domain of a different structure (four-stranded antiparallel $\beta$-sheet, stacked on two $\alpha$-helices) that does not harbor a Quercetin-pocket. The presence of HuR and FUS on MTs after their fusion to MBD and their interactions with specific mRNAs onto MTs was confirmed beforehand (*Figure 4c*). Ten molecules were tested, of which five hits (F2, C3, C8, A3, P1) and five negative controls (P2, F11, F5, D4, C12), and none of them did significantly affect mRNA:HuR or mRNA:FUS interaction scores (*Figure 4b*). Hence, the five selected hits specifically target mRNA interactions with YB-1.

## In vitro validation of targeting the Quercetin-pocket

An in vitro validation of the binding of the above selected compounds to the Quercetin-pocket was also conducted using NMR spectroscopy. Here, ligand binding was detected via changes of protein resonances in 2D $^1$H-$^{15}$N spectra upon ligand addition using a 1:4 protein:ligand ratio. However, from the 40 molecules selected by the in silico approach and tested with the MT bench assay, only 25 (of which 8 CTRLs) were amenable to solution NMR studies (15 of them presented solubility issues, notably the in cellulo hit A3). Analysis of the chemical shift data show significant CSPs for the pocket residues for 15 of the molecules being tested including all the 11 putative hits identified with the MT bench assay in cells (except for A3). Average pocket CSPs are reported in *Figure 2—source data 1* along with MT bench scores. F1, F4, F6, F8, and F9 did not significantly decrease mRNA:YB-1 interactions in cells even though the amplitude of the CSPs in vitro indicate a significant binding to the pocket. Parameters related to the cellular context in which the MT bench assays were performed such as half-life of compounds, potential off-target interactions, membrane permeability and/or selectivity toward the YB-1 targeted pocket most likely account for the discrepancy between cellular and in vitro data.

## Data mining of ligand-induced CSPs reveals P1 selectivity

To analyze compound selectivity towards the Quercetin-pocket and make a parallel between in vitro and cellular results, we implemented an in-depth structural analysis based on the NMR chemical shift perturbations using data mining techniques. The aim is to (i) examine how these ligands target the Quercetin-pocket differently and (ii) identify key residues relevant to differential ligand selectivity. To

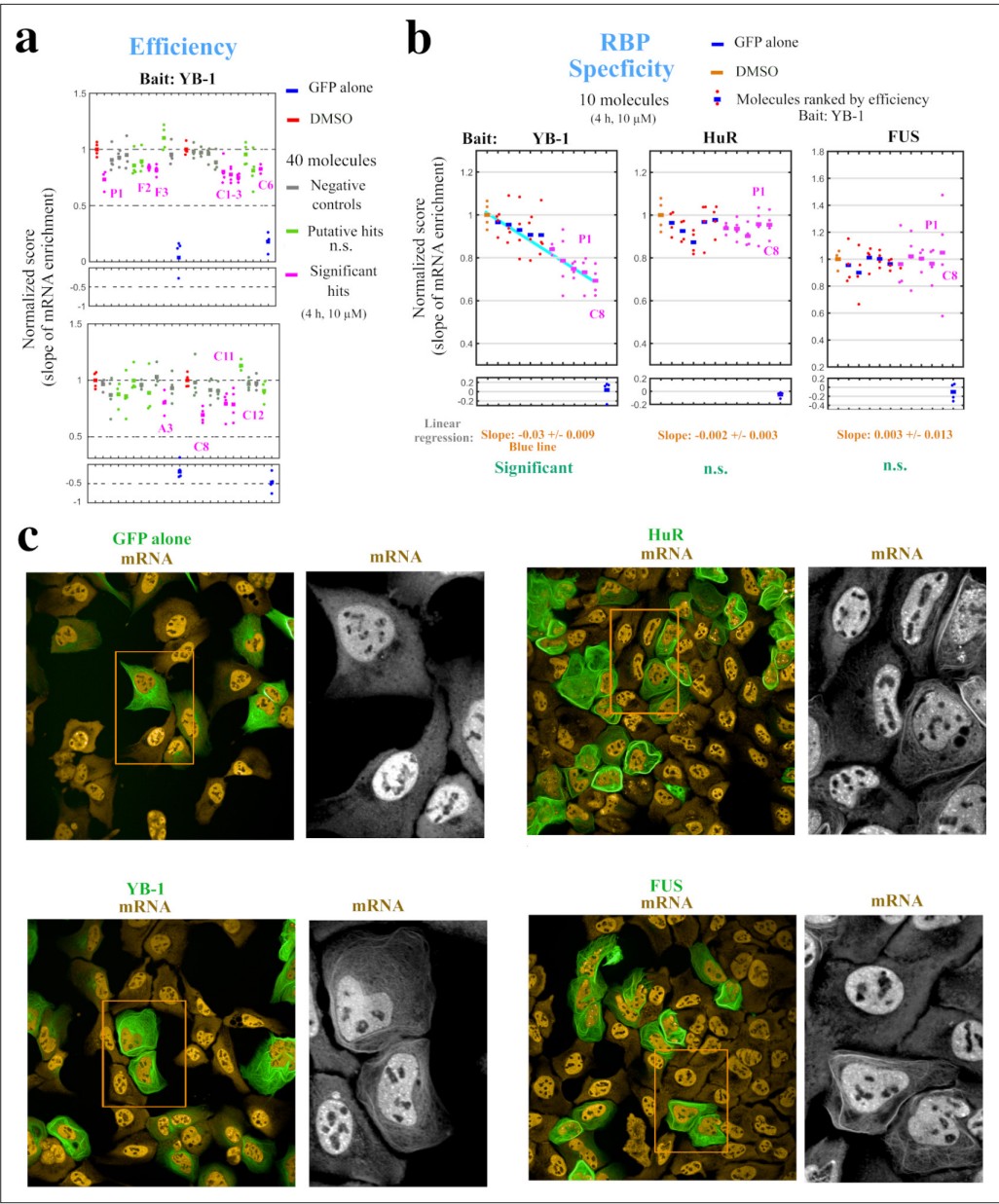

**Figure 4.** Hits identified with the MT bench assay target selectively mRNA:YB-1 interactions. (**a**) Normalized mRNA enrichment slope measured with MT bench in U2OS cells expressing YB-1 as bait after 4 hr exposure to the selected 40 molecules at 10 μM. The slope of the mRNA enrichment on MT versus bait expression was measured in quadruplet in two 96-well plates (20 molecules *per* plate). Each plot represents a plate; negative controls (grey), non-significant (n.s.) putative hits (green), significant hits (magenta), DMSO control (red), GFP control (blue). Compounds were selected as significant hits (magenta) when $p < 0.05$ according to a paired $t$-test relative to DMSO controls. (**b**) The specificity of the molecules to YB-1 in (**a**) was tested against two other RBP baits, HuR and FUS. Left panel: five negative controls (blue, red) and five significant hits (magenta), from (**a**) were selected and ordered on the $x$-axis according to their efficiency to affect mRNA:YB-1 interactions according to (**a**). Their interaction score is shown on the $y$-axis. The blue line represents the decreasing slope with 95% confidence intervals. Middle and right panels: five negative controls and five significant hits for YB-1, from (**a**), tested against HuR and FUS, respectively. The same ordering of compounds in $x$-axis from the left panel was used and a non-significant slope was measured (n.s.). No significant hit was detected for both RBPs. P1 and C8 are labeled in all three panels. (**c**) Images representing the expression and localization of 3 different RBPs used as bait (MBP-GFP-RBP) and a negative control (MDB-GFP) in U2OS cells. The 3 baits used here are HuR (top right panel), YB-1 (lower left panel), and FUS (lower right panel). All the RBP baits tested were successfully detected on MTs (green) and efficiently brought mRNAs onto MTs (orange).

*Figure 4 continued on next page*

*Figure 4 continued*

The online version of this article includes the following source data for figure 4:

**Source data 1.** Normalized slope values for *Figure 4b*.

this end, a principal component analyses (PCA) was performed on a 15 by 20 2D matrix (denoted $A$) and its transpose $A^T$ built from the average CSPs ($\Delta\delta_{avg}$) of 15 ligands and 20 YB-1 residues; analysis details are described in Appendix 4 section I and results are illustrated in *Figure 5*. Here, linear dimensionality reduction using singular value decomposition of the data to project it to a lower dimensional space aims to reveal hidden simplified structures. As a result, the accumulative contribution ratio of the first six principal components (PCs, linear combinations of the CSPs of YB-1(CSD)

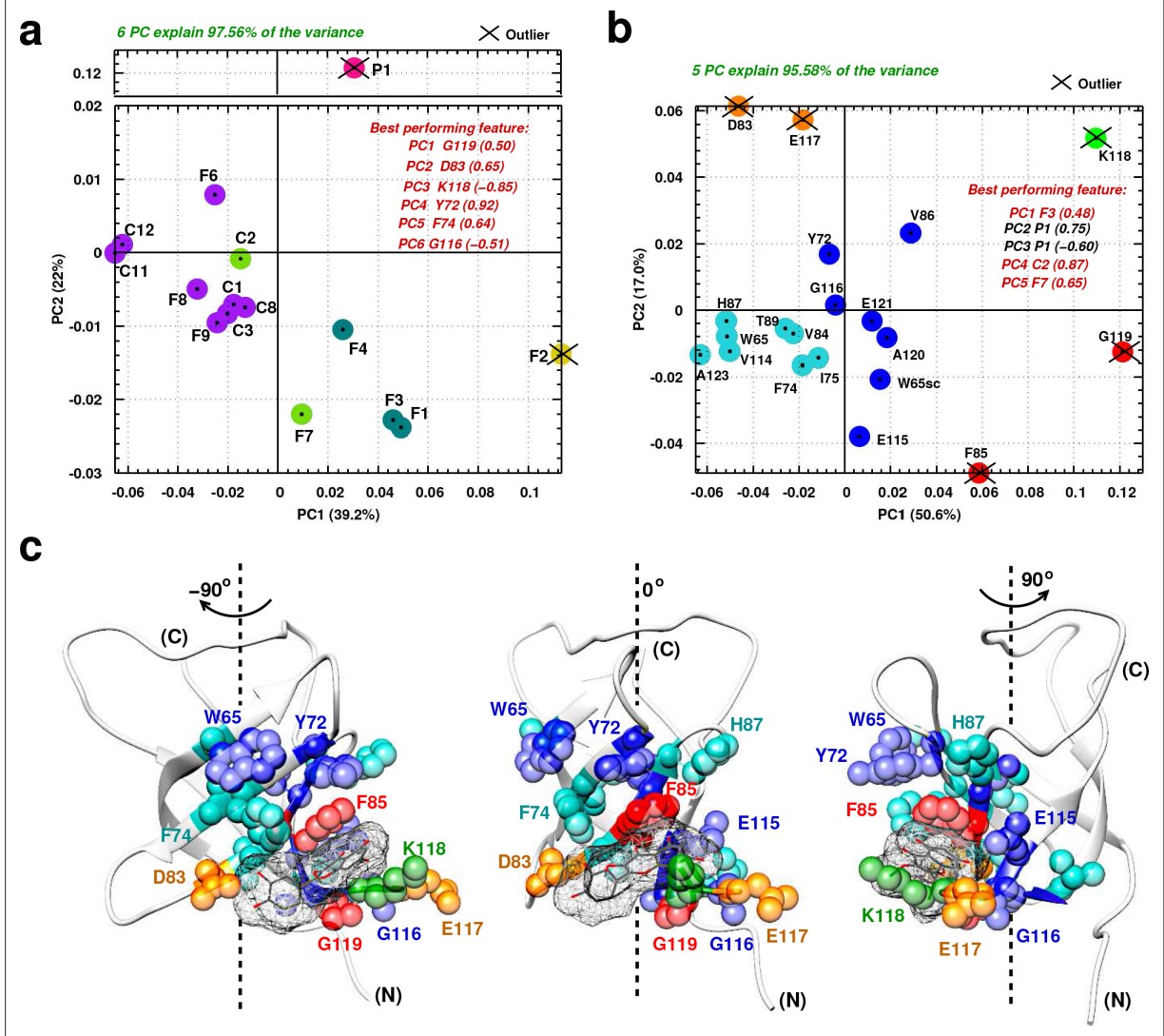

**Figure 5.** Principal component analysis of ligand-induced CSPs from ¹H-¹⁵N NMR spectra of ¹⁵N-labeled YB-1. (a, b) A 2D scatter plot of PCA results obtained on matrix $A$, where $A$ = [15 ligands x 20 residues], and its transpose $A^T$ [20 residues x 15 ligands]. The scatter plot shows PCA projected vectors for different ligands (**a**) and residues (**b**) on the first and second PC dimension (PC1 and PC2). Different colors indicate five clusters found for the 6 (**a**) and 5 (**b**) PCs by cluster analysis. The best performing features by PC with absolute highest loading are also listed. Outliers based on SPE and Hoteling's T² tests are also indicated. (**c**) Projection of PCA results on the 3D structure of a YB-1:Ligand complex. The protein is represented in light grey cartoon; the ligand (here F3, taken as an example) in dark grey sticks and a mesh surface. Residues showing significant loadings from PCA are represented in spheres and colored based on the colors of the clusters formed in (**b**). (**N**) and (**C**) indicate the N-and C-terminal, respectively.

The online version of this article includes the following figure supplement(s) for figure 5:

**Figure supplement 1.** Correlation-matrix-based hierarchical clustering of NMR ligand-induced CSPs.

residues) for matrix $A$ was 0.97, meaning that these 6 dominant dimensions are likely to describe most contributions to the signal. YB-1 residues that correspond to the best performing features by PC are K118, G119, G116, Y72, F74, and D83 (*Figure 5a*). On the ligand side, two outliers were detected, P1 (magenta) and F2 (yellow), meaning that each of these 2 ligands target the Quercetin-pocket differently compared to the rest of the ligands. PC results also reveals five clusters where each gathers group of similarly-acting ligands (all clusters are color coded in *Figure 5*). Consistent with their identification as outliers, P1 and F2 each belong to a cluster of their own.

A similar analysis can be performed with the matrix transpose $A^T$. In this case, five PCs explain 95% of the variance with an accumulative ratio of 0.95 (*Figure 5b*). The ligands that correspond to the best performing features by PC are P1, F3, C2, and F7. In addition, clustering of the PC results shows five clusters of similarly-affected residues, and where the five PC outliers detected are grouped into three sets of YB-1 residues: K118 (green), [F85;G119] (red), and [D83;E117] (orange).

In order to explain the observed residue-related results, a direct comparison with the binding modes obtained by MD was essential. *Figure 5c* shows a color-coded projection of $A^T$ PCA results on a 3D structure of a YB-1:Ligand complex, where F3 is taken as an example. The five residue outliers (forming the first three clusters) are residues involved in the direct binding of YB-1 to the ligands (green, red, orange). The two remaining clusters with higher populations (blue and cyan) divide the binding site residues into three sets: (i) direct neighbors of residues making direct interactions with the ligands (such as V86 and G116 that are direct neighbors of outliers F85 and K118), (ii) residues located in the vicinity of the pocket that interact with residues that bind the ligand (such as Y72 and W65sc that are related to F85 via $\pi - \pi$ stacking) and (iii) residues located further away (such as T89, I75, and V114); and where in these cases the observed CSP is due to indirect binding or structural rearrangement.

Cross validation of PCA results between matrix $A$ and its transpose yields striking observations. For instance, P1 is distinguished as the only ligand to appear as an outlier in $A$ (magenta) and as the best performing ligand, with highest loadings, in two of the five dominant PCs in $A^T$. Looking from the residue side of things, K118 (green), G119 (red), and D83 (orange) are highly distinguished as best performing residues in $A$ and outliers in $A^T$. Hence, according to the PCA analysis, the higher selective binding of P1 to the Quercetin-pocket, compared to the other tested ligands, may be due to the interaction of P1 with central (F85, K118, G119) and peripheral (D83, E117) residues.

## FDA-approved P1 binds YB-1(CSD) with a Kd of 6 µM in silico and in vitro

Since the FDA-approved P1 was found to have the highest selectivity toward the Quercetin-pocket compared to flavonoid- and chalcone-like molecules, we then further scrutinized the interaction of P1 with YB-1. First, in order to assess whether P1 presents one or multiple binding modes, we conducted 2D $^1$H-$^{15}$N NMR titrations. The superposition of the titration spectra produced a straight line (and not curved plots) which is indicative of a single binding mode (*Figure 6—figure supplement 1a*). The multiple binding in general produces curved plots, because the secondary interactions will almost always have different effects on the chemical shifts than the primary interaction (*Craven et al., 1996*; *Williamson, 2013*). Second, Saturation Transfer Difference (STD) NMR (*Mayer and Meyer, 2001*) was carried out to investigate the binding of P1 to YB-1 (*Figure 6—figure supplement 1b*). The obtained epitope mapping of P1 illustrates which chemical moieties of the ligand are key for molecular recognition in the binding site and allowed us to unequivocally orient P1 in the Quercetin-pocket. The resulting epitope mapping confirms the 3D structure of P1 bound to YB-1 obtained by MD simulations.

Next, the binding mode was assessed by an extended 200 ns MD simulation. The free energy landscape of YB-1:P1 complex sampled from MD (*Figure 6a*) shows a local energy minima over a large free energy space (deep basins, dark blue) indicating that the protein structure has become a minimum energy structure during the simulation period. The overlay of several structures sampled and extracted from the basins shows stability and the same binding mode within fluctuations. The interaction energies ($\Delta$H) averaged over the simulation between P1 and the pocket residues along with its electrostatic (Coul) and Lennard-Jones (LJ) contributions show that ~80% of the binding is due to van der Waals (vdW) contacts (–85.9 kJ.mol– 1 for LJ vs –20.8 kJ.mol– 1 for Coul). The highest contributing residues to the binding are F85 with –25.5 kJ.mol– 1 and K118 with –19.8 kJ.mol– 1, where P1 is retained in the Quercetin-pocket by a hydrophobic sandwich (*Figure 6b and c*). A strong

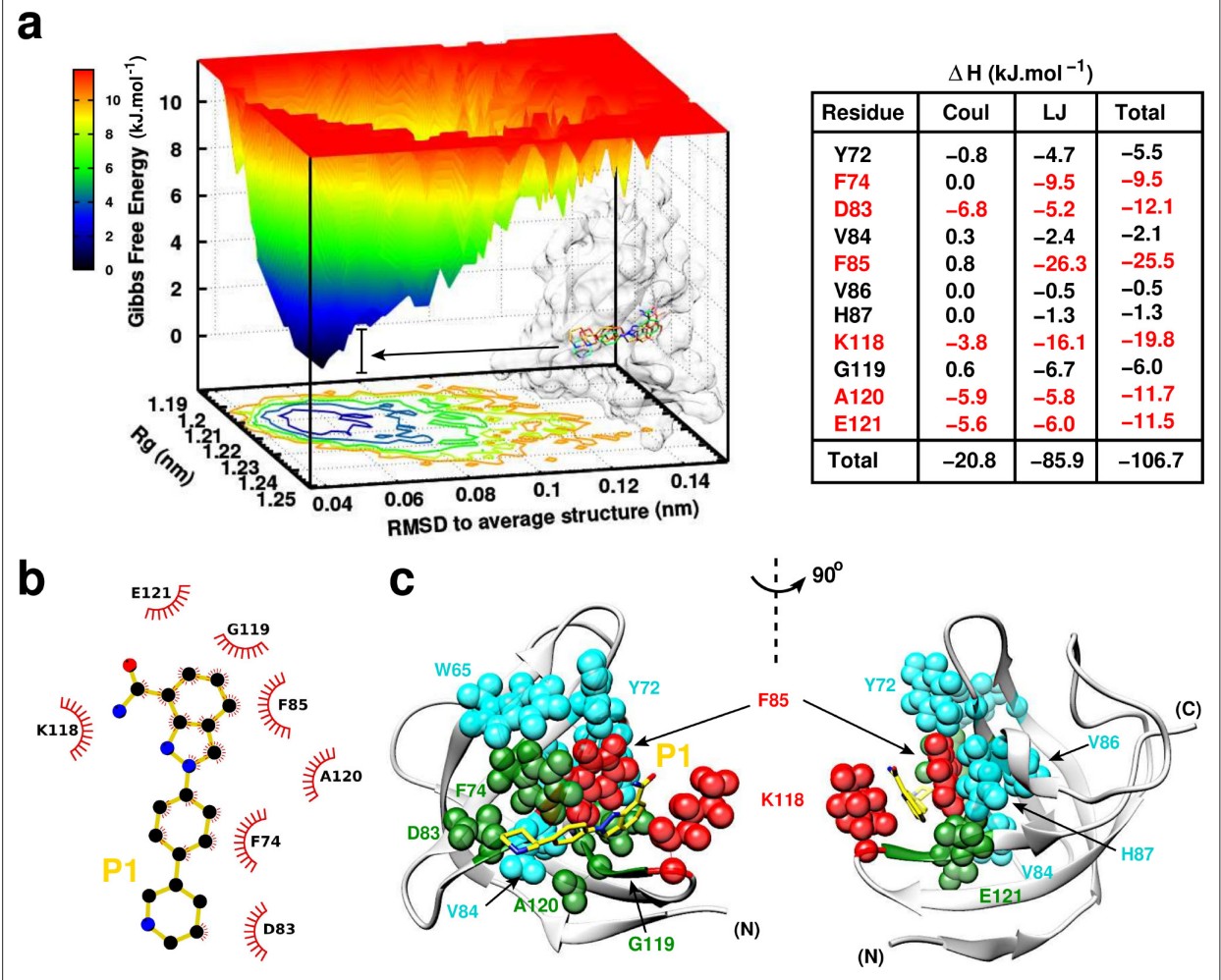

**Figure 6.** P1:YB-1 binding mode assessed by MD simulations. (**a**) Left panel: Free Energy Landscape (FEL) of P1 in complex with YB-1 computed from 200 ns MD simulations. FEL is represented using two structural reaction coordinates: the radius of gyration of the system and the root mean square deviation (RMSD) with respect to the average structure. The zero energy is at 0 kJ.mol−1 and corresponds to the most stable conformational states. The free energy scale highlights energy differences (0–12 kJ.mol− 1) relative to the global minimum. Radius of gyration and RMSD values are reported in nm. The 3D representation shows 'valleys' of low-free energy corresponding to the metastable conformational states of the system, and 'hills' account for the energetic barriers connecting these states. The free energy surface is also projected as a 2D 'contour plot' on $x$- and $y$-axis. The inset shows an overlay of several conformational states sampled from two low energy wells (indicated by black arrows); the protein is shown as transparent light grey surface. Right panel: Interaction energy contribution (**ΔH**) of the residues implicated in the binding, along with its Coulomb (Coul) and Lennard-Jones (LJ) contributions, averaged over 200 ns of MD simulation with variant of fluctuations being ±1.6 kJ.mol− 1. The most contributing residues are marked in red. (**b**) 2D interaction diagram between P1 (gold) and YB-1 residues. (**c**) 3D representation of the zero-energy complex. The protein is represented in light grey cartoon, P1 in gold sticks. Residues implicated in the binding and showing significant CSP in NMR 2D ¹H-¹⁵N-SOFAST-HMQC spectrum (**Figure 7—figure supplement 1**, top left panel) are represented in spheres: residues with high interaction energy and/or high CSPs are in red, intermediate (green), lower (cyan). (**N**) and (**C**) indicate the N- and C-terminal, respectively.

The online version of this article includes the following figure supplement(s) for figure 6:

**Figure supplement 1.** NMR investigation of P1 binding to YB-1.

**Figure supplement 2.** Mapping the effect of F85A mutation on P1 binding to YB-1 by MD and NMR Mapping the effect of F85A mutation on P1 binding to YB-1 by MD and NMR.

**Figure supplement 3.** NMR investigation of P1 binding to LIN28(CSD) and HuR(RRM2).

$\pi - \pi$ stacking between the indazole ring and F85 from one side and strong vdW interactions with K118 from the opposite side. Moreover, P1 is also retained/pinned by E121 located at the right side of the pocket due to electrostatic and vdW interactions. The middle benzene ring engages in a perpendicular $\pi - \pi$ stacking with F74 (–9.5 kJ.mol– 1). In addition, the piperidine cycle of P1 is pinned by D83 located at the left side of the pocket.

Isothermal titration calorimetry (ITC) measurements were also conducted in order to determine the binding affinity of P1 to YB-1 (**Appendix 5—table 1**). Results indicate a binding free energy ($\Delta G_{bind}$) of –7.14 kcal.mol−1; which translates into a $K_d$ of    . These results are in line with the calculated $\Delta G_{bind}$ from ABFE simulations of –7.24 kcal.mol−1 (**Appendix 5—table 2**). In addition, both results (ITC and ABFE) show that the driving force for P1's association with its target is enthalpic, meaning an enthalpy-driven association. The observed low enthalpy and entropy values from simulations compared to ITC results are due to an underestimation of the $\pi - \pi$ stacking interactions involving aromatic and non-aromatic groups computed with a point charge force field. However, this did not affect the resulting calculated free binding energy which reproduces the experimental ITC value, within statistical errors.

Since, the key interaction in P1's binding to YB-1 is the strong $\pi - \pi$ stacking with F85, we next decided to change this residue into alanine. MD and NMR results show that YB-1 mutant F85A no longer interacts with P1 (**Figure 6—figure supplement 2a, b**, respectively). The structural investigation of MD results reveal that the binding pocket collapses due to F85A mutation since F85 maintained the aromatic side chains of residues H87, Y72, F74 and W65 along with K118 in place (**Figure 6—figure supplement 2c**). Finally, to ascertain the specificity of YB-1:P1 interactions in vitro, we tested whether P1 interacts with two other RBPs LIN28A(CSD) and HuR(RRM2) using 2D NMR. YB-1(CSD) is different in residue composition compared to the LIN28 family (**Moss and Tang, 2003**) (LIN28A and LIN28B), two other CSD proteins with a high degree of sequence homology with YB-1 in humans (~40% residue identities). LIN28(CSD) is structurally similar to YB-1(CSD) with few residue mismatches located in the

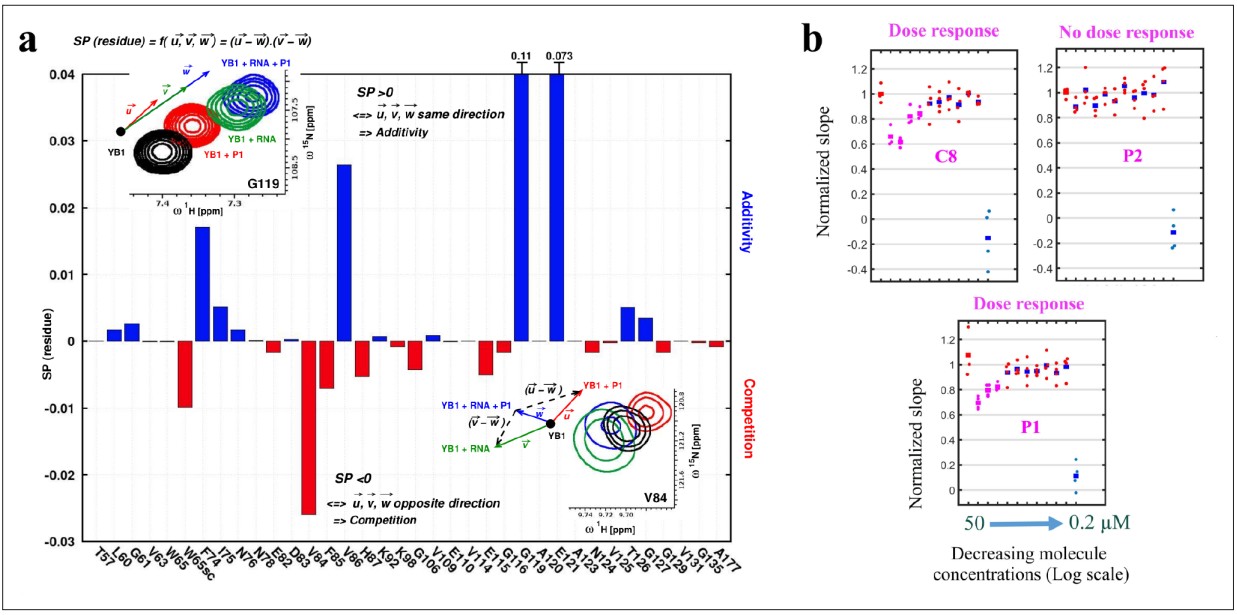

**Figure 7.** P1 interferes with RNA:YB-1 interactions in vitro and in cells. (**a**) Histogram showing the competitive behavior of P1 using $SP(residue)$ calculated based on CSPs extracted from ${}^1$H-${}^{15}$N-SOFAST-HMQC spectra of ${}^{15}$N-labeled YB-1 in the absence and/or presence of P1 and 5-nt long poly(C) RNA. The insets show a zoom-in on residue V84 (exhibiting competition) and G119 (exhibiting additivity) from overlaid NMR spectra of YB-1 alone (black) and in the presence of: P1 (red), RNA (green), and both P1 + RNA (blue); an illustration of the associated displacement vectors $\vec{w}$, $\vec{v}$ and $\vec{u}$ (same color code as NMR), relative to YB-1 alone (large black dot), and the pair vectors ($\vec{v}$ - $\vec{w}$) and ($\vec{u}$ - $\vec{w}$) (black dashed arrows) on which the scalar product was calculated are also indicated. (**b**) Dose response plots of P1, C8, and P2 in U2OS cells expressing YB-1 as bait (MT bench) following 4 hr exposure at decreasing concentrations from 50 to 0.2 µM (quadruplicate in 96-well plate).

The online version of this article includes the following source data and figure supplement(s) for figure 7:

**Source data 1.** Normalized slope values for **Figure 7b**.

**Figure supplement 1.** Mapping Ligand/RNA competition on binding YB-1 using NMR.

**Figure supplement 2.** Investigating C8 binding to YB-1 in the presence of RNA by MD and NMR.

Quercetin-pocket (mainly Y72/F, G116/S, E117/A, and A120/L) that generate a different structural rearrangement of the side chains. HuR(RRM2) is an RNA-binding domain of a different structure (four-stranded antiparallel $\beta$-sheet, stacked on two $\alpha$-helices) with no Quercetin-pocket. NMR results show that some residues in the LIN28(CSD) pocket were experiencing CSPs but to a significantly lesser extent than YB-1(CSD). In addition, we noticed that CSPs in LIN28(CSD) residues are located outside of the Quercetin-pocket, demonstrating a weak and nonspecific binding to LIN28(CSD) (*Figure 6—figure supplement 3a*). Regarding P1 binding to HuR(RMM2), no relevant interaction was detected (*Figure 6—figure supplement 3b*).

To sum up, P1 is found to bind YB-1(CSD) via vdW interactions (mostly) with a high affinity (K $_d$ of ~, measured in vitro (ITC) and in silico (ABFE)) and with a certain specificity when compared to the two other RBPs used in this study, LIN28(CSD) and HuR(RRM2). Here, MD simulations provided a resolved atomistic picture of the binding mode and revealed the inhibition mechanism. Furthermore, MD and NMR analysis of the F85A YB-1 mutant in complex with P1 emphasize on the leading role of F85 in targeting the Quercetin-pocket.

## P1 interferes with RNA:YB-1 interactions in vitro and in cells

To put to the test whether P1 can significantly interfere with the binding of YB-1 to mRNAs in vitro, the CSPs of YB-1 residues located in the Quercetin-pocket in the presence of 5-nt long poly(C) RNA with or without P1 was analyzed. To calculate the CSPs induced by P1 in the presence of RNA in solution, the YB-1:RNA spectrum was used as a reference (see inset in *Figure 7a*). Results show additional CSPs associated with P1, apart from those due to RNA:YB-1 interactions, which indicates the presence of both YB-1:RNA and YB-1:P1 complexes in solution. In order to analyze and assess the competitive behavior of P1, we thus considered in detail the chemical shift variations from different $^1$H-$^{15}$N-SOFAST-HMQC NMR spectra of $^{15}$N-labeled YB-1 in the absence and/or presence of P1 and/or 5-nt long poly(C) RNA. For this, the ligand's ability to compete with RNA on YB-1 binding was evaluated using the scalar product of pair displacement vectors, here denoted $SP(residue)$ (*Figure 7a*; a detailed description of the analysis is provided in Appendix 4 section II). These vectors correspond to the chemical displacement induced after adding the ligand ($\vec{u}$), RNA ($\vec{v}$) and both RNA +ligand ($\vec{w}$) to YB-1. Hence, YB-1 residues that display chemical shifts moving in opposite directions in the presence of P1 and RNA compared to RNA alone will have a negative $SP$ such as G119, and residues displaying chemical shifts moving in the same direction will have a positive $SP$ such as V84 (see insets in *Figure 7a*). These observations can be translated into a 'competition' or 'additive' regime for $SP < 0$ and $SP > 0$, respectively. Among the residues showing competition, several are directly involved in the interaction with RNA such as W65sc, V84, F85, and E82. Their observed CSPs have negative $SP$ values, which is what is expected in a competition for binding. Similar competing behavior was also observed for C8, another hit used here as a positive control (*Figure 7—figure supplement 1* (middle panels) and 7), while P2 (Olaparib, another PARP inhibitor used here as a negative control) showed no effect (*Figure 7—figure supplement 1*, lower panels). Although in vitro results show that P1 and C8 can interfere with the binding of CSD to RNA, the Quercetin-pocket represents only a part of the RNA:CSD interface which involves at least four consecutive nucleotides. However, in agreement with the results of the MT bench assay, many RBPs have to compete with each other to gain access to mRNAs in cells. A compound that interferes slightly with the RNA:YB-1 interface may thus dramatically shift the balance toward RBP competitors and lead to an apparent decrease in the affinity of YB-1 for mRNA in cells which cannot be observed in vitro. To further test whether P1 significantly decrease the affinity of YB-1 for mRNA in cells owing to its binding to YB-1(CSD) and not to its PARP inhibition activity, we planned a series of experiments. First, the dose responses of P1, C8, and P2 were analyzed (*Figure 7b*). The mRNA enrichment slope was measured in quadruplicate in 96-well plates. Results show that P1 and C8 but not P2 displayed a classical dose response with a critical concentration of about $10$ M, consistent with a low μM range affinity for YB-1 as calculated by free energy simulations and/or by ITC (K $_d$ for P1 by ABFE simulations and ITC; and K $_d$ for C8 by ABFE simulations). In addition P3, P4, P5, three others PARP inhibitors did not affect mRNA:YB-1 interactions in our first screen at $10$ M (*Figure 4b*) and no significant CSPs in YB-1 residues were detected in the presence of P2, P3 or P4 in vitro (*Figure 2—source data 1*). Only P5 was found to bind to the Quercetin-pocket of YB-1 but with a significantly lower affinity than P1, which was also confirmed by ABFE simulations (estimated K $_d$; but it can range between 30 and $90$ M considering the 0.5 kcal.mol−1 error on the computed value).

Therefore, any potential effect resulting from the inhibition of PARP by P1 on mRNA:YB-1 interactions in cells can be ruled out. Altogether, the results obtained in silico by MD simulations, in vitro by NMR and, in cells with the MT bench assay point toward the ability of P1 to compete with mRNA for binding YB-1 at $\mu$M concentrations.

## May P1 affect YB-1 cellular functions related to mRNAs?

Finding functional cellular assays that would reveal a phenotype specific to a general RBP (such as YB-1) is not an easy task, and it is even more difficult with YB-1 since it binds non-specifically to most mRNAs (*Singh et al., 2015*) as shown from CLIP analysis (*Wu et al., 2015*). In addition, YB-1 is an abundant protein in cancer cells. In HeLa cells, the cellular model chosen here, YB-1 abundance is ranked 248 among all proteins with 1.7 million copies *per* cell (*Nagaraj et al., 2011*).

In order to reveal a phenotype related to the interaction between YB-1 and mRNA, we decided to expose HeLa cells to elevated P1 concentrations (increase from 20 to $100\,\mu$M) during 2 hr. Below $50\,\mu$M, no change in YB-1 and mRNA distribution could be noticed in the cytoplasm of HeLa cells (*Figure 8—figure supplement 1a, b*). Above $200\,\mu$M, cells underwent massive death. However, at $100\,\mu$M of P1, even if nonspecific activity cannot be avoided so close to the toxic concentration threshold, we detected the presence of YB-1-rich granules in the cytoplasm with two different anti-YB-1 antibodies (*Figure 8a* and *Figure 8—figure supplement 1c*). YB-1 granules appeared only in few cells treated with P1, which may reflect a cell phase dependency, but repeatedly in many different and independent experiments. While YB-1 granules can be considered nonspecific stress granules related to cellular stress, they were distinct from stress granules (SGs) triggered by Arsenite (*Khong et al., 2017*; *Bounedjah et al., 2014*), a potent and widely used inducer of SGs in cells (large cytoplasmic SG in *Figure 8a*, upper right panel). Consistent with a decreased affinity of YB-1 for mRNAs at high P1 concentrations, mRNAs were poorly recruited in YB-1 granules compared to SGs formed in the presence of Arsenite. This is illustrated in the scatter plots of *Figure 8a* for SG (Arsenite, orange scatter plot) and YB-1 granules (P1, blue scatter plot). None of the other PARP-1 inhibitors led to the formation of YB-1-rich granules (*Figure 8a*, *Figure 8—figure supplement 1a, b*). In addition, YB-3, which shares an identical CSD with YB-1, is also significantly recruited in these granules. As a control, HuR which is not a target of P1 is recruited to a lesser extent than YB-3 (*Figure 8—figure supplement 1d*).

We then considered whether P1 affects YB-1 function related to mRNA translation. As YB-1 binds to most non polysomal mRNAs (*Singh et al., 2015*), YB-1 may regulate the overall translation rates in cells by controlling the switch from a polysomal state (active) to a non polysomal state (dormant). When mRNAs are blocked in a non polysomal state, cellular translation rates should decrease. Accordingly, in a recent work, we showed that YB-1 unwinds non polysomal mRNAs in a way that facilitates the translation from dormant to active state. In agreement with another report in myeloma cells (*Bommert et al., 2013*), we also showed that decreasing the expression of YB-1 reduces mRNA translation in HeLa cells (*Budkina et al., 2021*). Hence, we tested whether P1 may interfere with mRNA translation. In a previous report, but after long PARP inhibitor treatment (72 hr), a decrease in translation level was measured by puromycin incorporation because of the activation of PARP-1 by small nucleolar RNAs (snoRNAs) in the nucleolus (*Kim et al., 2019*). To limit this bias, we chose to measure mRNA translation after short P1 treatment (2 hr) and compared the results obtained with P1 with two other PARP-1 inhibitors that do not target the Quercetin-pocket, P2 and P3. The incorporation of puromycin to nascent peptide chains during translation is significantly reduced in cells treated with P1 but not P2 and P3 at concentrations as low as $2.5\,\mu$M (*Figure 8—figure supplement 2*). We then decided to directly probe whether the inhibition of mRNA translation detected with P1 was YB-1-dependent. For this HeLa cells were pre-treated with two different siRNA to decrease endogenous YB-1 levels and with siNEG as a negative control. Cells were then exposed to indicated molecules ($10\,\mu$M, 2 hr) and briefly exposed with puromycin before fixation to estimate global mRNA transition at the single-cell level. In cells treated with two different siRNAs targeting YB-1, we observed that P1 did not significantly impair mRNA translation whereas a significant decrease in mRNA translation was measured with the negative control siRNA, (siNEG)-treated cells (*Figure 8b*, *Figure 8—figure supplement 3*). In contrast, P2 had no measurable impact on mRNA translation in both siRNA- and siNEG-treated cells.

Since YB-1 expression is associated to elevated cancer cell proliferation (*Evdokimova et al., 2009*; *Alkrekshi et al., 2021*), we also probed whether P1 reduces the cell number in a YB-1-dependent

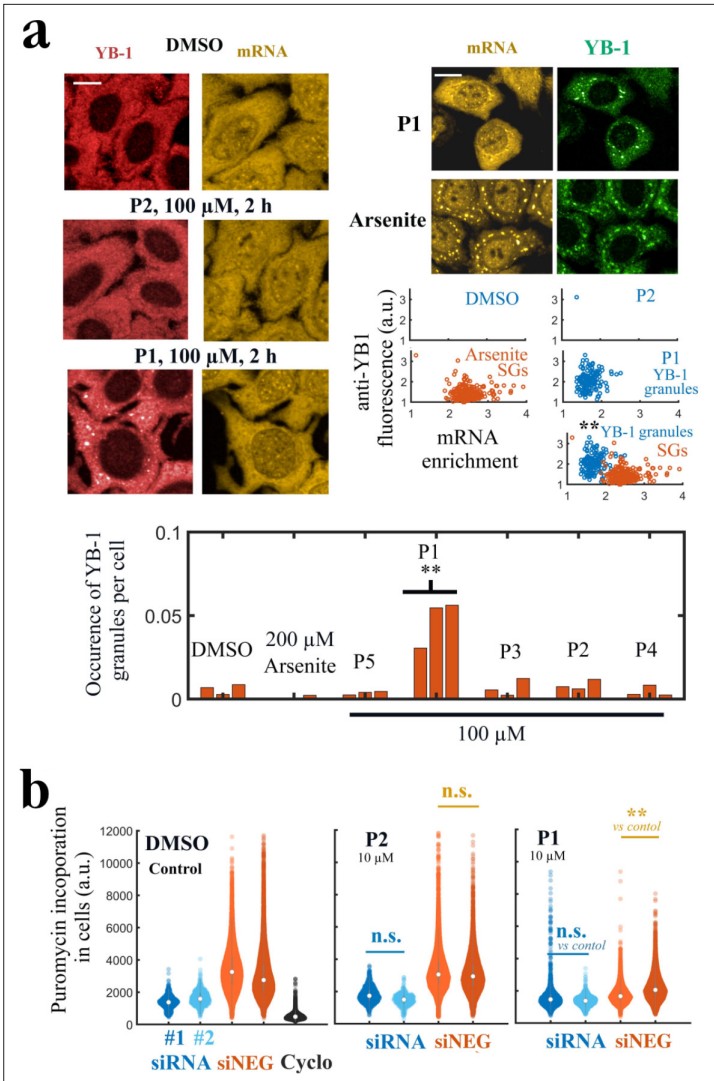

**Figure 8.** P1 alters cellular YB-1 functions in HeLa cells independently of PARP activity. (**a**) YB-1 distribution in cells exposed to P1. Upper left panels: HeLa cells treated with P1 and P2 at 100 µM for 2 hr. Upper right panels: Images of cells exposed to P1 and Arsenite. The corresponding scatter plots indicate a difference in mRNA-enrichment of stress granules (Arsenite, orange scatter plot) and YB-1 granules (P1, blue scatter plot). Lower panel: Occurrence of YB-1-rich granules in cells exposed to indicated treatment for 2 hr. **, $p<0.01$; $t$-test with two tails (triplicates). Scale bars: 20 µm. (**b**) HeLa cells pretreated with two different siRNAs to decrease endogenous YB-1 levels or siNEG (negative control), and then exposed to indicated molecules (10 µM, 2 h) followed by a brief exposure to puromycin before fixation to estimate global mRNA transition at the single cell level. **, $p<0.01$; $t$-test with two tails.

The online version of this article includes the following source data and figure supplement(s) for figure 8:

**Figure supplement 1.** Cytoplasmic YB-1-rich granules in HeLa cells at high P1 concentrations (≥20 µM).

**Figure supplement 2.** Monitoring puromycin incorporation in HeLa cells.

**Figure supplement 3.** Detection of puromycin incorporation in HeLa cells.

**Figure supplement 4.** P1 increases the separation distance between cells when comparing YB-1-rich with YB-1-poor HeLa cells.

**Figure supplement 4—source data 1.** Anti-YB-1 fluorescence intensity values for *Figure 8—figure supplement 4b*.

**Figure supplement 4—source data 2.** Data points of the calculated distance between closest neighbors for *Figure 8—figure supplement 4c*.

*Figure 8 continued on next page*

*Figure 8 continued*

**Figure supplement 5.** Compared to other PARP-1 inhibitors, P1 further decreases the number of cells *per* well in YB-1 rich cells.

**Figure supplement 5—source data 1.** Number of cells per well data points for *Figure 8—figure supplement 5*.

manner. To this end, we measured the number of HeLa cells plated at low density in 12-well plates after having decreased, or not, YB-1 levels with siRNA. In addition, we used P2 and P3 as negative controls as they inhibit PARP-1 but do not target YB-1 like P1. *Figure 8—figure supplement 5* shows that all PARP-1 inhibitors decrease the cell number, albeit to a higher extent with P3. However, both P2 and P3 further decrease the number of cells in siRNA-treated cells compared to siNEG-treated cells (with significant differences at 5 µM), which may be due to reduced resistance to stress when YB-1 expression is decreased. In contrast, P1 rather further decreases the number of cells in siNEG-treated cells when YB-1 levels are high (non-significant variations but opposite to those observed with P2 and P3). The separation distance between cells also decreases significantly in YB-1-poor cells (siRNA) treated with P1 compared to siNEG-treated cells, in contrast to P2 and P3 (*Figure 8—figure supplement 4*). A short distance of separation between cells may be due to colony formation when cells were plated at low density and allowed to grow for 48 hr. Therefore, while we may have expected a higher sensitivity of cells to P1 when the YB-1 expression is low, in contrast, P1 seems to further decrease cell number when YB-1 level is high, which may be due to a gain of toxic or cytostatic function, notably a decreased translation rate as observed in *Figure 8b*. However, further analyses need to be undertaken to document this point.

Therefore, the appearance of YB-1-rich granules and the inhibition of the YB-1-dependent mRNA translation in HeLa cells are consistent with P1 interfering with mRNA:YB-1 interactions. To which extent P1 may affect YB-1-related functions in cells remains to be investigated in details.

## Discussion

In this study, we introduce an integrative approach that leads to the identification of several effective YB-1 inhibitors in the low micromolar range selected computationally and validated in vitro by NMR spectroscopy and in cells using the MT bench assay. Here, the MT bench was adapted to score small molecules targeting RBP interactions with endogenous mRNA in cells. The MT bench assays can notably fill the gap between in vitro and functional assays by probing whether the interaction of a selected RBP with mRNAs is affected in a cellular context but not that of other RBPs. Our results validate the reliability of the MT bench assay in detecting and scoring YB-1 interactions with mRNA in 96-well plates (SSMD >8, *Figure 3c*).

Using a rationally designed large-scale computational approach, 22 potentially effective compounds (along with 18 CTRL) targeting a druggable pocket located at the YB-1(CSD):RNA interface (the Quercetin-pocket) were selected to be tested (*Figures 1 and 2*). An in vitro structural validation using protein-based NMR data, which is necessary to ascertain their capability in targeting the Quercetin-pocket, was also conducted when possible. The MT bench assay revealed that 11 out of the 22 selected hit compounds significantly decrease the interaction of YB-1 with mRNA in cells. In contrast, when 5 hits were tested with two other RBPs (FUS and HuR), no decrease of RPIs was observed (*Figure 4*). Here, endogenous poly(A)-tailed mRNA was used as bait to detect mRNA:YB-1 interactions since YB-1 is a general mRNP factor in the cytoplasm. New developments may enable to target RBPs interacting with specific RNA (mRNA encoding a specific gene, ncRNA, etc.), which may be helpful for the challenging issue of developing molecules that would target an interaction between a specific RNA and a specific RBP.

All of the molecules selected in this study are multi-aromatic ring systems that are sandwiched in the Quercetin-pocket. Besides their common anchoring key residues, F85 and K118, other interactions with neighboring residues, from both sides of the pocket, are needed to stabilize the ligands in the binding pocket, as clearly indicated by the PCA analysis (*Figure 5*). The PCA analysis also identified key residues implicated in the high selectivity of P1 toward the Quercetin-pocket that can thus be used to rationally optimize our leads. As for the computational approach implemented in this study, the validity of its predictive potential was challenged in vitro and in cells. In vitro, 15 predicted hits out of 17 were confirmed to bind YB-1 in the targeted pocket by NMR (yielding 88% success rate); and in

cells, 11 out of 22 were found to inhibit RNA:YB-1 interactions (50% success rate, while only C6 of the 18 negative controls emerged as a significant hit). Other factors may play a role in yielding negative results for predicted hits such as off-targets and cell permeability, which is precisely the point of using the MT bench assay. Here, we managed to balance computational accuracy and cost by using the point-charge force field CHARMM. However, further efforts can be applied to optimize the computational approach by using advanced multipolar and polarizable force fields in order to improve the ranking and reduce the errors (*Gresh et al., 2015*). Owing to the rapid feedback between atomistic, chemical, structural, and cellular data integrated here, our ligand screening strategy for RPI inhibitors may also be refined.

Apart from the FDA-approved inhibitors, these molecules belong to the flavonoid family with known anti-inflammatory or anti-tumor activity in humans (*Panche et al., 2016*), except for C3, C4, C8-C10, and the A series for which no known activity was reported. However, given the many biological processes on which these compounds interfere (*Panche et al., 2016*) and their numerous targets, their selectivity may be doubtful. Nevertheless, a rational optimization of these Chalcone- and Flavonol-like molecules guided by quantum chemical calculations and relative free energy simulations may increase their affinity and selectivity to YB-1.

The top lead, on which we focused the rest of our analysis, is P1, an FDA-approved PARP-1 inhibitor. Based on an exhaustive structural analysis, P1 displays the highest selectivity by targeting key residues from all sides of the pocket via mostly vdW interactions (*Figures 5 and 6*). Moreover, it represents a clear specificity to mRNA:YB-1(CSD) interactions when compared to LIN28(CSD) and HuR(RRM2) in vitro. Altogether, the results obtained in silico by MD simulations, in vitro by NMR, and in cells with the MT bench point toward an inhibition of mRNA:YB-1 interaction by P1 at low micromolar concentrations, which is consistent with the moderate affinity of P1 for the YB-1 Quercetin-pocket ($K_d$ 6 µM). In addition, functional assays show a global decrease in YB-1-related mRNA translation, cell proliferation, and the appearance of YB-1-rich granules in HeLa cells treated with P1. Given the positive regulation of mRNA translation and the negative regulation of stress granule assembly exerted by YB-1 in HeLa cells (*Budkina et al., 2021*), these results thus did not exclude the possibility that P1 may target YB-1 functions related to mRNA in HeLa cells.

As YB-1 is a secondary target of P1 and finding secondary targets of FDA-approved PARP-1 inhibitors has been a recent concern due to their different indications and multiple adverse effects, especially P1 (*LaFargue et al., 2019*; *Knezevic et al., 2016*), we may consider whether impairing YB-1 function in cells may provide a rational explanation for the observed adverse effects of P1. For instance, the pronounced hematological adverse effects, particularly thrombocytopenia (*LaFargue et al., 2019*) may be explained by the role YB-1 in megakaryocyte versus erythroid differentiation (*Bhullar and Sollars, 2011*). However, more data are needed to explore the putative involvement of YB-1 in P1 adverse effects.

In summary, we have developed an integrative approach to specifically target RPIs in cells with small molecules. While the data are promising for RNA:YB-1 interactions and provide a first proof of concept, we would like to stress out that this is not yet sufficient to assert that this approach could be successful with all RBPs. Separate studies are needed to validate the MT bench for other RBPs. YB-1 is an 'ideal' target because it has a single cold-shock domain and a druggable pocket, which may not be the case for other RBPs. In addition, many RNPs harbor several RNA-binding domains, which may reduce the sensitivity of our method when a specific domain is targeted by small molecules because the other domains would contribute to the binding to mRNAs. However, a single RNA-binding domain may be isolated and used as bait for the MT bench assay to overcome this obstacle. Developing molecules that would target a specific domain may be sufficient, to modulate the biological function exerted by the full length protein.

## Materials and methods

### Computational methods

#### System setup and molecular dynamics simulations

For this study, the following systems were considered for MD simulations: WT YB-1 protein (apo form), WT YB-1:RNA(C5) complex (holo form), YB-1:Ligand complexes and YB-1-F85A mutant in complex with P1 Niraparib.

The starting 3D coordinates of YB-1 CSD used the NMR solution structure PDB code 1H95 (*Kloks et al., 2002*) and the YB-1:RNA(C5) complex was constructed using as a template the crystal structure of YB-1 cold-shock domain in complex with UCAACU (PDB ID 5YTX (*Yang et al., 2019*), resolution 1.55 Å). The protein sequence is of 85 amino acid in length going from A45 to G129. The YB-1:Ligand complexes were generated, in a next step, using as a building block for docking screened ligands, an MD sampled open-state of YB-1. As for the mutant F85A bound to P1, it was generated from the lowest energy state MD refined WT YB-1 structure in complex with P1.

All MD simulations were carried out with GROMACS software package version 2018.2 (*Abraham et al., 2015*) using the additive force field CHARMM27 for proteins (*MacKerell et al., 1998*) and nucleic acids (*Hart et al., 2012*) with periodic boundary conditions. Ligands parameters were obtained using SwissParam (*Zoete et al., 2011*) which provides topology and parameters for small organic molecules compatible with the CHARMM all atoms force field. The protonation states of the residues were adjusted to pH 7.6 (pH used in our NMR experiments). The systems were centered and solvated in a triclinic box of TIP3P (*Jorgensen et al., 1983*) water model with 1.4 nm distance between the boundary of the box and the system in question. A [KCl] of $100\,\text{mM}$ was used and counter-ions were added to neutralize the system. Each system was first energy minimized using 50,000 steps of steepest descent, then heated to 298 K at constant volume for 500 ps and equilibrated in the NPT ensemble at $p$ = 1 atm for 500 ps which was followed by 10 or 200 ns of NPT production run depending on the aim of the computational protocol. A 10 ns of MD production run was used for MD pose refinement of YB-1/ligand complexes in order to allow protein rearrangement upon ligand binding. These refined poses were then used for subsequent $\Delta G$ calculations, respectively. The long MD simulations were used to study the evolution as a function of time of YB-1 apo state, YB-1:RNA and YB-1:P1 complex. The Velocity Rescaling (*Bussi et al., 2007*) (with $\tau$ = 0.1 ps) and Parrinello-Rahman (*Parrinello and Rahman, 1981*) methods were used for temperature and pressure control, respectively. The equations of motion were propagated with the leap-frog (*Van Gunsteren and Berendsen, 1988*) algorithm and the time step was $\Delta t$ = 2 fs. The particle mesh Ewald (PME) (*Darden et al., 1993*; *Essmann et al., 1995*) method was used for electrostatic interactions, with grid spacing of 1.6 Å, a relative tolerance of 10−5, an interpolation order of 4 for long-range electrostatics, and a cutoff of 14 Å together with a 12 Å switching threshold for LJ interactions. All bonds with hydrogen atoms were constrained with LINCS (*Hess, 2008*).

## Virtual screening

The virtual screening part of the in silico approach is divided into 3 parts: (i) the pharmacophore-based screening; (ii) the automated blind docking of FDA-approved drugs; and (iii) the physico-chemical and purchasability filters applied. The procedure is detailed in Appendix 2.

## Statistical mechanics-based filter

After a visual inspection of the docking results (*Fischer et al., 2021*) and carefully selecting the docked poses of the ligands, these docked complexes were reassessed using a dynamic view. Compared to the static docking approach, here the complex is simulated in presence of an explicit solvent, where real enthalpic and entropic contributions are taken into account. Short MD simulations of 10 ns are sufficient to refine the docked pose and to check the stability of the ligand in the binding site. Ligands that left the binding site during this 10 ns were discarded, and only ligands that stayed in the binding site were considered for the following.

Knowing that the ligand can be destabilized by the water molecules interacting with its unbound side (on the water accessible surface), we defined a first observable that allows us to estimate if the ligand will reside in the pocket for a longer time. This can be translated into the ability of the water network interacting with the bound ligand to extract it from the binding site. In order to estimate this, we calculated the difference between ligand-protein ($\Delta H_{LP}$) and ligand-water ($\Delta H_{LW}$) interaction energies along the MD simulation. This enthalpic observable ($\Delta\Delta H_{(LPvsLW)}$=$H_{LP}$-$H_{LW}$) was then averaged over the 10 ns of MD, in order to score the ligand by its preference to reside in the pocket $\Delta\Delta H_{(LPvsLW)} < 0$ or in the solvent $\Delta\Delta H_{(LPvsLW)}>0$. Standard deviations ($\sigma$) fluctuated between 2 and 7 kcal.mol−1 depending on the size and the complexity of the ligand.

Another observable denoted $c$ was also defined based on the ability of the ligand to make more than one interatomic contact *per* interacting atom. This was expressed as the ratio between (i) the

total number of unique atomic contacts made between the ligand and the pocket residues (ii) and the number of ligand atoms in contact. Ligand-Pocket interatomic contacts were calculated from the MD refined structure. Interatomic contacts are defined based on their vdW radius, and englobe polar and nonpolar interactions.

With these two observables we were able to define a simple weighted scoring function in order to determine the best-performing ligands:

$$S = \sum_{i=1}^{2} w_i(Obs_i) \tag{1}$$

with $w_c = 4$ and $w_{\Delta\Delta H} = -1$ introduced to differently weight the two observables. This scoring function is rather more qualitative than quantitative and allowed us to sort our ligands into potential hits or not. Ligands with $S > 0$ were considered as potential hits. However, ligands with $S < 0$ were also considered as potential hits as long $\sigma$ for $\Delta\Delta H_{(LPvsLW)}$ was higher than the actual average value. This filter proved to be computationally efficient (short 10 ns MD) for screening a large number of ligands.

## Absolute binding free energy simulations

Using a non-physical thermodynamic cycle, the absolute binding free energy of YB-1:ligand complexes ($\Delta G_{bind}$) was calculated as sum of free energy change of formation of protein-ligand complex formation ($\Delta G_{complex}$) and the free energy of desolvating the ligand ($\Delta G_{solv}$); to which an analytical correction term for adding restrains on the decoupled ligand was added ligand ($\Delta G_r$).

The free energy difference between two end states was estimated using the Bennett Acceptance Ratio (BAR) (*Bennett, 1976*). Here, the ratio of weighted average of Hamiltonian difference of two given states is calculated using multiple intermediate states defined by the coupling parameter $\lambda$ to monitor the alchemical transformation. Hence, the Hamiltonians for the states were determined by combined Hamiltonians for the end states A and B. The linear relationship $H_\lambda = H_A + \lambda(H_B - H_A)$ leads to a Hamiltonian representing states A and B, respectively. The initial and final states are defined as A ($\lambda = 0$) where the ligand is absent and B ($\lambda = 1$) where the ligand is fully grown.

In these equilibrium simulations, the system is coupled/decoupled by applying a scaling parameter $\lambda$ to the nonbonded interactions, which switches between the initial ($\lambda = 0$, state A) and final state ($\lambda = 1$, state B). The interval $0 < \lambda < 1$ was divided into 40 equally spaced windows. First, the LJ interactions with soft-core potentials (*Beutler et al., 1994*) are fully grown, followed by the electrostatics in the presence of the full vdW interactions, thereby avoiding the need for soft-core electrostatic potentials. For each of these steps, the system was re-equilibrated for 500 ps followed by 2 ns of dynamics in the NPT ensemble during which information was accumulated. For the solvation free energies, the system was-re-equilibrated in the NPT ensemble for 200 ps and information was accumulated for 1 ns.

Auxiliary restraints were used to prevent the ligand from leaving the binding site when the native ligand-receptor interactions were turned off alchemically. These restraints restrict both the position and the orientation of the ligands and are defined relative to the receptor. This free energy cost can be evaluated analytically using *Equation 2* (*Boresch et al., 2003*):

$$\Delta G_r^{VBA,0} = -RT \ln \left[ \frac{8\pi^2 V^0}{(r_{\alpha A,0}^2 \sin\theta_{A,0} \sin\theta_{B,0}} \frac{(K_r K_{\theta A} K_{\theta B} K_{\Phi A} K_{\Phi B} K_{\Phi C})^{1/2})}{(2\pi RT)^3} \right] \tag{2}$$

,where $R$ refers to ideal gas constant, $T$ is temperature in Kelvin, $V^0$ is standard system volume for 1 molar concentration, $r_0$ is reference distance for restraints, $\theta_A$ and $\theta_B$ are reference angles for restraints, $K_x$ refers to strength constant of distance ($r_0$), two angles ($\theta_A, \theta_B$), and three dihedrals ($\Phi_A, \Phi_B, \Phi_C$). The ligands were restrained by means of one distance and force constant of 1000 kcal.mol$^{-1}$.nm$^{-2}$, two angles, and three dihedral harmonic potentials with force constant of 10 kcal.mol$^{-1}$.rad$^{-2}$.

## Protein expression and purification

The recombinant His6-tagged YB-1 fragment ($^1$Met-$^{180}$Gly) from the human full-length YB-1 was first cloned into the pET22b expression vector at NdeI/XhoI sites. BL21 (DE3) competent *E. coli* cells were transformed with the constructed plasmid pET22b-YB1_1–180 and grown at 37 in 1 L 2YT-ampicillin medium (non-labeled proteins) or in minimal medium M9 supplemented with 15NH4Cl (labeled proteins). The protein expression was induced by IPTG 1 mM added at OD $_{600nm}$=0.7. The culture was

grown at 37 for 4 hr and cells were harvested and washed with 20 mL of cold 20 mM Tris-HCl buffer, pH 7.6, containing 100 mM KCl. The cell pellet was resuspended in 10 mL of buffer A (20 mM Tris-HCl, pH 7.6, 2 M KCl, 0.5 mM DTT, 1 mM PMSF, 10 mM Imidazole, and EDTA-free protease inhibitor Cocktail (Roche)) and cells were disrupted by sonication on ice (Bioblock Vibracell sonicator, model 72412). The cell lysate was centrifuged at 4 for 30 min at 150,000×g in a TL100 Beckman centrifuge.

The YB-1 (1-180) protein fragment was purified following the manufacturer's recommendations (Qiagen). Briefly, the supernatant was incubated for 2hr at 4 with Ni2+ - NTA-agarose (Qiagen) (20 mg of proteins/ml of resin) pre-equilibrated in buffer A. The resin was then washed extensively with buffer A containing 10 mM imidazole and by reducing progressively the KCl concentrations (from 2 M till 0.5 M). The elution of the protein was performed by adding 250 mM imidazole in buffer A and fractions were pooled and diluted 25 x with 20 mM Tris-HCl, pH 7.6, 0.5 mM DTT, 1 mM PMSF in order to incubate them with protease-free RNase A (Thermo Scientific) for 90 min at room temperature. The protein pool was finally re-purified on the same conditions as described above, dialyzed against 20 mM Tris-HCl, pH 7.6, 0.5 M KCl and stored at -80°C.

Site-directed mutagenesis of the human YB-1 coding gene was carried out directly on the pET22b-YB-1_1–180 expression plasmid by using the "Quikchange II XL site-directed mutagenesis kit" from Stratagene and appropriate oligonucleotides (Eurofins Genomics). The introduced mutation (F85A) was validated by DNA sequencing (Eurofins Genomics). Overexpression and purification of YB-1 (1–180 aa) mutant F85A were performed by following the same protocol detailed above.

## NMR spectroscopy

All NMR experiments, protein- or ligand-based were performed at 600 MHz on a Bruker AVIII HD spectrometer equipped with a triple-resonance cryoprobe. All samples were prepared in a final volume of 200 L using 3 mm diameter tubes. NMR data were processed with Topspin 4.0 (Bruker). Assignment of $^1$H and $^{15}$N chemical shifts of YB-1 (1–180 aa) was retrieved from our previous study (*Kretov et al., 2019*) and from the results obtained by *Zhang et al., 2020*.

## Characterization of the purchased compounds, solubility and stability assessment

All compounds were purchased from Molport, except for F3 and F6 that were purchased from SigmaAldrich, C2, C5, C6, C7, C8 from CarboSynth and C3 from Ambinter, and had purity >97% (compound IDs and supplier codes are provided in *Appendix 5—table 3*; their chemical structures are displayed in *Figure 2*). Marvin was used for drawing, displaying and characterizing chemical structures, Marvin version 19.16.0, ChemAxon (https://www.chemaxon.com). Purity and solubility were verified by acquiring $^1$H NMR spectra for each compound dissolved in 100% DMSO-d6. Next compound solubility was checked in aqueous buffer (50 mM phosphate buffer at pH 6.8 and 298 K), to ensure a 1 mM final concentration, by measuring peak integrals from $^1$H-NMR spectra compared to an internal reference. All occurring peak variations due to instability or solubility issues were monitored over time within a 48 hr time period by acquiring 1D $^1$H NMR spectra at regular intervals (*Sreeramulu et al., 2020*). Instability issues due to fast degradation were mostly observed for some of the flavonoids.

From the 40 molecules purchased, 15 represented solubility issues and thus were not amenable for NMR studies with YB-1. These molecules are: F10, C4, C5, C7, C9, C10, A1 to A8 and D3. Hydrophobic buffers such as MOPS can be used instead of phosphate buffer to solubilize these ligands, however this exceeds the scope of this paper. A prior testing of how the YB-1 (1–180) fragment will behave in a different buffer environment should be evaluated beforehand.

## Chemical shift perturbation analysis and titration

Free, ligand- and RNA-bound protein samples were prepared in NMR buffer (20 mM Tris, pH 7.6, containing 100 mM KCl, 10% D2O) supplemented with SUPERase·in RNase Inhibitors (Eurogentec) for RNA samples. All the protein-ligand samples were prepared in a 1:4 protein:ligand ratio. Typically, the final protein and ligand concentrations were 50 M and 200 M respectively, and the final volume was 200 L. However, due to low solubility a ratio of 1:1.8 and 1:1.25 was used for F7 and F9, respectively. For samples containing RNA, the protein:RNA ratio was 1:1.2 and the protein:RNA:ligand ratio for competition experiments was 1:1.2:4. A DMSO-d6 percentage of 2% was maintained in all experiments. For P1 titration essay, a 50 M protein solution was incubated with increasing ligand

concentrations from 10 to $1000\,M$, where a constant percentage of DMSO-d6 (2%) was maintained. The number of titration points was 16. Ligand and RNA binding to YB-1 were investigated using 2D $^1$H-$^{15}$N SOFAST-HMQC (*Schanda and Brutscher, 2005*) at 298 K. The number of dummy scans and scans was respectively set to 16 and 256. Data were acquired with 2048 points along the direct dimension and with 128 $t_1$ increments with a relaxation delay of 0.2 s. Spectral widths were set to 12.5 ppm (centered at 4.7 ppm) in the $^1$H direction and 30 ppm (centered at 118 ppm) in the $^{15}$N dimension. Shaped pulse length and power were set by considering an amide $^1$H bandwidth of 4.5 ppm and a chemical shift offset of 8.5 ppm.

Ligand binding was followed by CSP analysis for which the weighted average chemical shift values were calculated and normalized according to *Equation 3*:

$$\Delta\delta_{avg} = \sqrt{0.5[\Delta\delta_H^2 + (0.14\Delta\delta_N)^2]} \tag{3}$$

where $\Delta\delta_{avg}$ is the average CSP at a given ratio, $\Delta\delta_H$ and $\Delta\delta_N$ are the chemical shift changes in the $^1$H and $^{15}$N dimension, respectively (*Williamson, 2013*). The NMR data analysis and interpretation approaches implemented and adapted in the purpose of this article are presented in full detail in Appendix 4. These include three data mining techniques: principal component analysis (PCA), correlation-matrix-based hierarchical clustering and scalar similarity measure. Even though the results of the correlation-matrix-based hierarchical clustering performed here are in line with the PCA, it was less sensitive and informative for ligand selectivity (*Figure 5—figure supplement 1* and Appendix 4 Section I).

## STD experiment and P1 ligand mapping

P1 ligand resonances were assigned through 2D $^1$H COSY, $^1$H-$^{13}$C-HSQC and $^1$H-$^{13}$C-HMBC spectra acquired on a 2 mM P1 solution in 50 mM phosphate buffer (pH 6.8) and 100 mM KCl at 298 K.

STD experiments were acquired on a YB-1:P1 sample prepared in a 50-fold excess of ligand ($500\,M$) at 283 K using a pseudo-2D Bruker pulse scheme (stddiffesgp.3) with excitation sculpting (*Hwang and Shaka, 1995*) for water suppression and a spinlock to suppress protein signals. The number of dummy scans and scans was respectively 32 and 1024. On-resonance irradiation was applied on one of the protein methyl resonance arising at –0.53 ppm, where no signal coming from the ligand is observed. The off-resonance carrier was set to 40 ppm, where no protein signals are visible. Selective presaturation of the protein was achieved by a cascade of 50ms Gaussian-shaped pulses (*Bauer et al., 1984*) corresponding to a total saturation time of 2 s.

Bruker AU program 'stdspli'" was used to process data. Integrals corresponding to the reference spectrum off-resonance spectrum ($I_{off}$) and to the difference spectrum between off- and on-resonance ($I_{diff}$) were extracted and used to calculate the fractional STD ($A_{STD}$) and the STD amplification factor $STD_{AF}$ (*Mayer and Meyer, 2001*) using *Equations 4 and 5*, respectively:

$$A_{STD} = \frac{I_{diff}}{I_{off}} = \frac{I_{off} - I_{on}}{I_{off}} \tag{4}$$

and

$$STD_{AF} = A_{STD}\frac{[L]_T}{[E]_T} \tag{5}$$

where $[L]_T$ and $[E]_T$ are the total ligand and protein concentrations, respectively. Relative STD percentages were derived by normalizing all STD integrals against the highest one obtained (assigned to a value of 100 %).

## Isothermal titration Calorimetry measurements of YB-1/P1 binding

ITC experiments were carried out at 25 with a MicroCal PEAQ-ITC isothermal titration calorimeter (Malvern Instruments). The protein sample was dialyzed against the ITC buffer (20 mM Tris-HCl, pH 7.6 containing 100 mM KCl and 2% DMSO). The protein concentration in the microcalorimeter cell (0.2 mL) was fixed at $14\,M$. 26 injections of 1.5 µL of P1 at $200\,M$ (resuspended in ITC buffer) were carried out at 90 s intervals, with stirring at 650 rpm and a reference power set at 11 µcal.s−1. In experiments with 5 nt-long poly(C) DNA (DNA(C5)), titration was carried out in the same buffer (without

DMSO), 18 injections of 2.0 µL of C5 at $200\,M$ and a reference power set at 5 µcal.s−1. Data were analyzed using the Microcal PEAQ-ITC Analysis Software and fitted using a one set-of-site binding model. All titrations were performed in triplicate.

## MT bench

### Cellular plate preparation for imagery

The MT bench assay was performed using bone osteosarcoma U2OS cells (ATCC HTB-96), a human cell line that was provided by O. Kepp (Gustave Roussy, Cell Biology Platform, Villejuif, France). U2OS cells were cultured at 37 in a humidified atmosphere with 5% CO2 in Dulbecco's modified Eagle's medium (DMEM, Life Technologies) supplemented with 10% FBS (Fetal Bovine Serum, Life Technologies) and 1% penicillin/streptomycin. The confluence of cells was verified every 5 days and cell were confirmed mycoplasma-free.

Cells were seeded on black 96-well plates cell carrier ultra (PerkinElmer) at a density of 16,000 cells *per* well using the liquid handler BRAVO from Agilent equipped with a 96-LT (Large Tips) head. After 24 hr incubation in a humidified incubator at 37 with 5% CO2, cells were transfected with $0.4\,g$ of indicated MBD-GFP-RFP plasmid for the positive control condition or with $0.2\,g$ of MBD-GFP plasmids by using 0.2 µL lipofectamine 2000 (Invitrogen) in optiMEM based on the optimization of the transfection conditions. The transfection complexes were prepared manually and transfection was done using the liquid handler BRAVO with specific transfection protocols depending on the type of plate that was prepared (Optimization, SSMD value, Hit Identification, IC50 determination).

Cells were treated in quadruplicate during 4 hr at 37 using 0.1% DMSO for the control wells at 10 µM of the different compounds for hit identification and with 10 concentrations ranging from 0.098µM to 50µM of the same compound for dose response assessment. The molecules were diluted in the culture medium, with a twofold serial dilution for the IC50 determination, and the treatment was made using the liquid handler BRAVO. A double fixation methanol/ParaFormAldehyde (PFA) was used to maintain the cellular protein cytoskeletal structure and allowing a good visualization of the microtubules. Cells were first fixed with ice-cold methanol 100% for 10 min at −20, washed with PBS and then further fixed with 4% PFA in PBS freshly prepared for 10 min at RT. After fixation, cells were incubated with oligo-dT-[Cy3], diluted in SSC 2 X, 1 mg.ml−1 yeast tRNA, 0.005% BSA, 10% dextran sulfate, 25% formamide, for 2 hr at 37 for RNA visualization. Wash steps were performed using 4 X and then 2 X SSC buffer (0.88% sodium citrate, 1.75% NaCl, pH 7.0). Cell nuclei were stained with DAPI (0.1 g.mL−1) for 5 min at RT. All the washing steps were performed with the Thermo Scientific Wellwash Versa Microplate Washer and the additions of the different solutions were done with the VIAFLO Electronic multichannel pipettes from Integra. Image acquisition was performed atomically with the Opera Phenix Plus High Content Screening System. Image analysis was performed with the HARMONY v4.8 software. Details on image acquisition and statistical analysis are provided in Appendix 3.

### RT-PCR analysis of RBP specificity

$10^6$ HEK cells were plated in six-well plates and transfected with the indicated plasmids with Lipofectamine 2000 reagent (Invitrogen). Twenty-four hr after transfection, cells were placed on ice for 30 min and lysed in 200 µL of lysis buffer (50 mM TrisHCl, pH 7.0, 50 mM NaCl, 1 mM EDTA, 0.05% sodium deoxycholate, 1% Triton X-100, 0.1% SDS, 1 mM PMSF, protease and RNAse inhibitors). Tubulin was purified from sheep brain as previously described (*Méphon-Gaspard et al., 2016*). Tubulin concentration was determined by spectrophotometry using an extinction coefficient of 1.2 mg−1.cm$^2$ at 278 nm. Tubulin polymerization was initiated by placing the ice-cold cuvette (1 cm light path) at 37 in a PTI QuantaMaster 2000–4 thermostated spectrofluorimeter. The kinetics of microtubule assembly was then immediately monitored by 90 angle light scattering at 370 nm. Microtubules were then taxol-stabilized (5µM taxol, 40µM tubulin).

Cell lysates were centrifuged at 20,000 g for 1 hr at 16 and the supernatant was collected. 10 µL of microtubule solution was added to 200 µL of cell supernatant, incubated for 15 min at 16 and centrifuged at 20,000 g for 30 min at 16. The microtubule pellet was resuspended in 100 µL of lysis buffer and again centrifuged at 20,000 g for 30 min. After discarding the supernatant, RNA was purified from the pellet with Tri-Reagent (Molecular Research Center, Inc) RNA quality and quantity was assessed by UV-spectrometry (nanodrop). RT-PCR reactions were performed using impromII Reverse transcriptase and GoTaq qPCR Master Mix on a 7500 Applied BiosytemsTM block. RNA quantification results

obtained with the microtubule pellet were compared to those obtained from the whole cell lysate. The oligo probes used for the RT-PCR analyzed are listed in *Appendix 5—table 4*.

For mRNA purification classical magnetic beads, HEK cells expressing indicated plasmids we lysed under conditions mentioned above. The purification assays were performed using Dynabeads Protein G Kit (Invitrogen) with anti-GFP antibody (monoclonal antibody, Invitrogen A11120, clone 3E6, IgG2a) in the same buffer used to isolate mRNA in a microtubule pellet, except the incubation time (here overnight in a cold room). RT-PCR analysis was performed as described above (results are listed in *Appendix 5—table 5*).

## Functional assays in HeLa cells

### Cell culture and transfection

HeLa cell lines (American Type Culture Collection, USA) were cultured at 37 in a humidified atmosphere with 5% $CO_2$ and maintained in the high glucose formulation of DMEM (Life Technologies) supplemented with penicillin G 100U.ml−1, streptomycin 100 µg.mL−1 and fetal bovine serum (FBS) 5% (10% for HeLa cells; Thermo-Fisher). The absence of mycoplasma was tested regularly to prevent any inference with the obtained results. The cell line identity was tested and authenticated.

The cells were grown in 24 or 96-well plates and transiently transfected with siRNA to decrease endogenous YB-1 levels with 2 different siRNAs siRNA-1: [sense 5'-(CCACGCAAUUACCAGCAAA) dTdT-3', anti-sense 5'-(UUUGCUGGUAAUUGCGUGG)dTdT-3']; siRNA-2 which targets the 3'UTR of YB-1 mRNA was used for the addback experiments [sense 5'-(GAUUGGAGCUGAAGACCUA)dTdT-3', anti-sense 5'-(UAGGUCUUCAGCUCCAAUC)dTdT-3']. The negative siRNA (1027310, Qiagen), siNEG, was applied in the same concentration as the two siRNAs. The mix of 1 g siRNA or siNEG in 300 µL optiMEM with 0.8 µL lipofectamine was left for 20 min at room temperature and added to cells for 3 hr, after that the solution was removed and the usual media was added to the well. Efficiency control was performed by immunofluorescence (*Figure 8—figure supplement 3a*). We obtained clusters of cells expressing endogenous YB-1 coexisting in the same sample with clusters of cells that displayed a significantly reduced expression of endogenous YB-1. Only the cells with a low YB-1 expression were retained for analysis (*Figure 8—figure supplement 3b*).

### Cellular translation assays

Hela cells treated with puromycin (10 µg.ml−1) for 10 min prior to fixation after washing out puromycin were fixed with 4% PAF for 30 min at 37 and subjected to immunoblotting using puromycin antibody (Merck, MABE343). For the negative control, cells were treated with cycloheximide (100 µg.mL−1) prior to the addition of puromycin. The anti-puromycin fluorescence in the cytoplasm was detected automatically using the Opera Phenix Plus High Content Screening System (PerkinElmer). The cytoplasm was detected automatically using the HARMONY v4.8 software.

### Detection of YB-1-rich granules

HeLa cells were subjected to indicated treatments for 2 hr. Cells were then fixed with methanol for 20 min at —20, followed with 4% paraformaldehyde for 30 min at 37. Immunofluorescence was performed with anti-YB-1–1 (rabbit polyclonal, Bethyl Laboratories, Montgomery, USA), anti-YB-1–2 (Anti-YBX1 antibody produced in rabbit, HPA040304, Sigma-Aldrich), Anti-YB-3 (Anti-YBX3 antibody produced in rabbit, HPA034838, Sigma-Aldrich) and Anti-HuR (antibody produced in mouse (3A2), 390600, Thermo Fisher Scientific). mRNA was detected by in situ hybridization as above-mentioned.

Quantifications were performed with Opera Phenix Plus High Content Screening System (Perkin-Elmer) in confocal mode. The HARMONY v4.8 software was used to detect and measure the number of cells having YB-1-rich granules, and the fluorescence intensity in the granules and in the cytoplasm for both and/or the number of SGs *per cell* (These values are directly accessible by selecting them in the 'spot analysis' parameters). The mRNA enrichment in YB-1-rich granules was measured by dividing the mean mRNA intensity in granules with the mean mRNA intensity in the cytoplasm.

### Cell number assay

HeLa cells were treated with siRNA-1 or siNEG overnight to decrease YB-1 level in most cells in siRNA-treated cells (*Figure 8—figure supplement 4b*). Then, Hela cells were plated at low density ($10^5$) in

12-well plates and treated-with indicated molecules for 48 hr. After cell fixation, cells were stained with anti-tubulin and DAPI. The number of cell was measured by an automatic detection of cell nuclei (HARMONY v4.8 software) as well as the distance between nearest neighbors.

## Acknowledgements

This project has received funding from the European Union's Horizon 2020 research and innovation program under the Marie Skłodowska-Curie grant agreement No 895024 (MSCA-IF-2019 to KEH). We thank the Centre Régional Informatique et d'Applications Numériques de Normandie (CRIANN), Project 2022010. This work was supported in part by the Région Ile-de-France (SESAME grant n°15013102), and Genopole (SATURNE grant 2020, HCS imager). This work was also supported by the INSERM PRI grant "RaPiD". We gratefully acknowledge the Genopole Evry, the University of Evry, and INSERM for constant support of the laboratory. We thank Pr. Olga I Lavrik and Dr. Maria V Sukhanova for the careful reading of the manuscript and valuable discussions. This paper is dedicated to the memory of our co-worker Guillaume Lambert who participated in our early work to decipher YB-1:RNA Interactions at the structural level.

## Additional information

### Competing interests

Nicolas Babault: Synsight has acquired a license for the "MT bench" patent (WO2016012451A1) concerning the industrial applications. Nicolas Babault is affiliated with SYNSIGHT. The author has no financial interests to declare. Pierrick Craveur: Synsight has acquired a license for the "MT bench" patent (WO2016012451A1) concerning the industrial applications. Pierrick Craveur is affiliated with SYNSIGHT. The author has no financial interests to declare. Hélène Henrie: Synsight has acquired a license for the "MT bench" patent (WO2016012451A1) concerning the industrial applications. Hélène Henrie is affiliated with SYNSIGHT. The author has no financial interests to declare. Cyril Bauvais: Synsight has acquired a license for the "MT bench" patent (WO2016012451A1) concerning the industrial applications. Cyril Bauvais is affiliated with SYNSIGHT. The author has no financial interests to declare. The other authors declare that no competing interests exist.

### Funding

| Funder | Grant reference number | Author |
| --- | --- | --- |
| Marie Sklodowska-Curie Actions | 895024 | Krystel El Hage |
| Genopole | SATURNE 2020 | David Pastré |

The funders had no role in study design, data collection and interpretation, or the decision to submit the work for publication.

### Author contributions

Krystel El Hage, Conceptualization, Data curation, Formal analysis, Funding acquisition, Investigation, Methodology, Project administration, Resources, Supervision, Validation, Visualization, Writing – original draft, Writing – review and editing, Developed, performed and analysed the computational part of the study (computational approach implementation, molecular dynamics simulations, free energy calculations), Performed and analysed NMR experiments; Nicolas Babault, Formal analysis, Investigation, Methodology, Supervised the MT bench work; analysed the MT bench data;, Implemented the HCS-based detection scheme; Olek Maciejak, Investigation, Methodology, Performed NMR experiments; Bénédicte Desforges, Investigation, Performed functional data assays; Pierrick Craveur, Formal analysis, Investigation, Methodology, Designed, performed and analysed the pharmacophore screen, Implemented the HCS-based detection scheme; Emilie Steiner, Investigation, Performed and analyzed the STD experiment; Juan Carlos Rengifo-Gonzalez, Formal analysis, Investigation, Methodology, Performed and analysed ITC experiments, Performed protein purification; Hélène Henrie, Investigation, Performed MT bench assay; Marie-Jeanne Clement, Investigation, Performed

NMR experiments; Vandana Joshi, Investigation, Performed molecular biology experiments; Ahmed Bouhss, Investigation, Performed protein purification; Liya Wang, Investigation, Performed MT bench assays; Cyril Bauvais, Resources; David Pastré, Conceptualization, Resources, Data curation, Formal analysis, Supervision, Funding acquisition, Validation, Investigation, Methodology, Writing – original draft, Project administration, Writing – review and editing, Supervised the MT bench work, Analysed the MT bench and the functional data, Implemented the HCS-based detection scheme

## Author ORCIDs
Krystel El Hage (iD) http://orcid.org/0000-0003-4837-3888
Olek Maciejak (iD) http://orcid.org/0000-0001-9594-9435
Pierrick Craveur (iD) http://orcid.org/0000-0001-9274-4944
Ahmed Bouhss (iD) http://orcid.org/0000-0002-6492-1429
Liya Wang (iD) http://orcid.org/0000-0001-7119-8665

## Decision letter and Author response
Decision letter https://doi.org/10.7554/eLife.80387.sa1
Author response https://doi.org/10.7554/eLife.80387.sa2

## Additional files

### Supplementary files
• MDAR checklist

### Data availability
All data are available within the article, supplementary siles and appendices, or available from the corresponding authors on reasonable request. Source data for Figures 2, 4d, 7b, Figure 3—figure supplement 3, Figure 8a, Figure 8—figure supplement 1b,c, Figure 8—figure supplement 4b,c, Appendix 5—table 1 and Appendix 5—figure 1 are also provided with the paper.

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

## Appendix 1

### I. Conformational study of YB-1 CSD in its unbound/free form

In order to study the dynamic behavior of YB-1 CSD and delignate the Quercetin-pocket, MD simulations were ran for 200 ns using as starting coordinates the NMR solution structure PDB ID 1H95 (*Kloks et al., 2002*). First, the free energy landscape (FEL) issued from the MD simulation was analyzed (*Figure 1—figure supplement 1a*). FEL is represented using two variables that reflect specific properties of the system and measure conformational variability: the radius of gyration and the root mean square deviation (RMSD) with respect to the average conformation; and the Gibbs free energy is estimated from the probability distribution of sampled populations. The zero energy is at 0 kJ.mol−1 and corresponds to the lowest energy conformational state (dark blue). A local energy minimum is observed over a large free energy space (deep basin, dark blue) indicating that these conformational ensembles are stable during the simulation period. The comparison of 2 structures extracted from the basin with a different radius of gyration show two different conformational states of a pocket (called here "the quercetin-pocket"): a closed state, where K118 is interacting with F85 (upper structure), and an open state where K118 is moved away from F85 into the solvent (bottom structure). This pocket is located at the third $\beta$-hairpin and somehow monitored by K118 and F85. To better understand the relation between these residues, we monitored the distance between the side chain N$_\zeta$ of K118 and C$_\gamma$ of F85 (red curve) along the simulation (*Figure 1—figure supplement 1b*, left panel). The probability distribution shows a first peak at ~3.7 Å which confirms a strong cation-$\pi$ interaction formed between the cationic side chain of K118 (NH$_{3+}$) and the electronegative benzene ring system of F85 (the cutoff being 6 Å), which is concomitant with a closed state pocket. The second is at 6.2 Å, meaning that the lysine is far away and thus the pocket exhibit an open state. The distance between the C$_\alpha$ of both residues was also monitored to see if the side chain movement is driven by the backbone. The distance probability distribution (black curve) shows one peak at ~7 Å, meaning that the observed cation-$\pi$ interaction is driven by K118 side chain movement and not by a backbone structural change of the U-turn. The higher probability of the open state (~75% of the time, compared to 25% for the closed state) is important to keep the pocket accessible for RNA binding. The RMSD of the pocket U-turn was also calculated for the Cα atoms of the protein (green), the β-sheet (black), and the U-turn (red) *Figure 1—figure supplement 1b*, right panel. Results show a high stability of the β-sheet of the CSD (<1 Å), a higher variation of protein Cα (~2.7 Å) owing to the flexibility of N- and C-terminal parts and U-turns. The RMSD of the pocket U-turn Cα atoms do not show high changes (broad peak 1–1.5 Å). In summary, these results show that the Quercetin-pocket presents an open and a closed state due to K118 side chain movement and that the opening mechanism is controlled by an electrostatic cation-$\pi$ interaction formed between the cationic side chain of K118 (NH3+) and the electronegative $\pi$-ring system of F85.

### II. Structural and energetic study of YB-1 CSD bound to C5 RNA

The YB-1 CSD bound to RNA was also investigated using MD simulations. To make a link with our experiments, a 5-nt long poly(C) RNA (C5) was used. The system was build using as a template the crystal structure of YB-1 CSD in complex with UCAACU (PDB ID 5YTX *Yang et al., 2019*), and simulations were run for 200 ns. The FEL plot shows low energy basins with a ~2 kJ.mol−1 difference between their local minima (*Figure 1—figure supplement 1a*). The structures extracted from the two observed wells show differences at the RNA extremities 5' and 3' which are highly flexible, and thus explains the two conformational ensembles. In order to identify the CSD residues implicated in the binding to C5 RNA, the interaction energy (ΔH) between individual residues and RNA was calculated and averaged along the MD simulation (*Figure 1—figure supplement 1b*). Results show key residues highly implicated in the binding such as K64, W65, Y72, F74, F85, H87, K118, and E121. The energy decomposition into Coulomb (Coul) and Lennard-Jones (LJ) contributions show that RNA C5 binds the CSD via electrostatic and vdW interactions equally. Three types of binding are observed: (i) purely electrostatic (K64 and E121), (ii) purely vdW via $\pi - \pi$ stacking (Y72, F74, F85), and (iii) both electrostatic and vdW (W65 and K118). *Figure 1—figure supplement 1c*, shows a 3D representation of the zero-energy complex on which a projection of significant CSPs obtained by NMR from 2D $^1$H-$^{15}$N-SOFAST-HMQC spectrum of $^{15}$N-labeled YB-1 in complex with C5 RNA is illustrated. The observed CSPs are in line with the binding mode from MD simulations, which in turn provides a resolved atomistic picture of the binding mechanism.

## III. Evolutionary conservation of YB-1 CSD

The evolutionary conservation of YB-1 CSD was evaluated by the ConSurf-DB server (*Ben Chorin et al., 2020*; *Goldenberg et al., 2009*). Calculations were done using default parameters and results are illustrated in *Appendix 1—figure 1*. The analysis show that: (a) the CSD in conserved in general; (b) the residues implicated in RNA-binding are the ones that are highly conserved (such as F74, F85, H87, and K118); (c) and the two residues that monitor the opening of the Quercetin-pocket (F85 and K118, who are also implicated in RNA-binding) are also highly conserved.

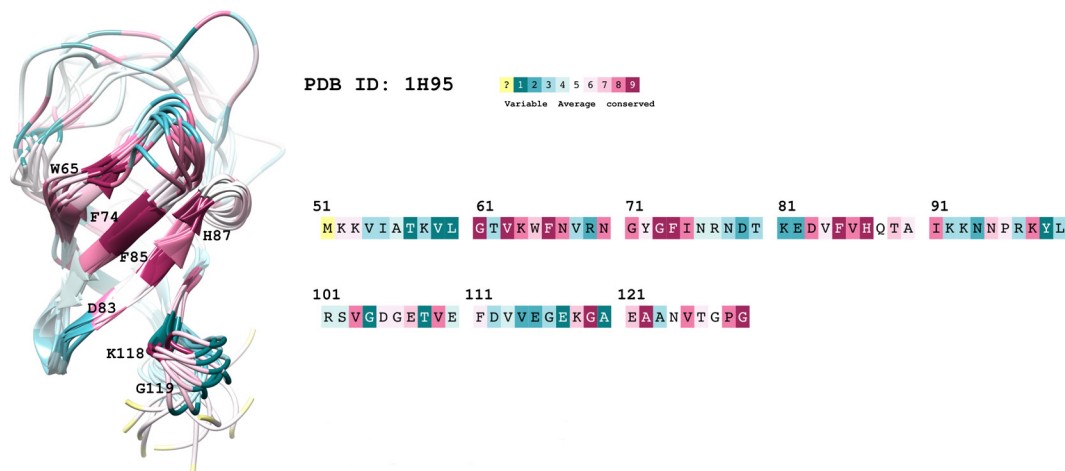

**Appendix 1—figure 1.** The conservation coloring profile from the ConSurf-DB repository, mapped onto the protein. The conservation coloring scale is shown on the top right conservation score varies from 1 to 9, where 1 corresponds to maximal variability and 9 to maximal conservation; "?" indicates insufficient data. Only highly conserved residues of interest are labeled on the 3D structure.

## Appendix 2

## Virtual screening details

### I.Pharmacophore-based screening

All docking computations were based on the minimized structure of the open-state conformation identified from the MD simulation of the apo form of the CSD of YB1. This structure was used to build two distinct pharmacophores. A first pharmacophore using the prediction of a 'pseudo ligand' in the binding site made by AutoSite (**Ravindranath and Sanner, 2016**) (see below), this approach is called 'Pocket-based'. And a second pharmacophore, called 'ligand-based', built from the 3D structure of YB-1 in complex with Quercetin (F1). This structure was obtained by docking F1 in the binding pocket of YB-1 followed by MD refinement and the target pocket was confirmed by NMR spectroscopy. Then we used both pharmacophores to virtually screen an in-house database composed of 208 million pharmacophores, representing the conformers of around 7.3 million distinct commercially available molecules from MolPort (https://molport.com).

From the 'pocket-based' screening, 249 distinct molecules were identified sharing at least 80% of their 3D pharmacophoric volume and sharing at least 5 pharmacophoric points. And the 'Quercetin-based' (or Ligand-based) screening allowed us to identify 407 distinct molecules sharing at least 60% of their 3D pharmacophoric volume and sharing at least 7 pharmacophoric points. None of the identified molecules were found in both virtual screening. To reduce these molecules to a final selection, we predicted ADME-T endpoints and computed molecular docking in the binding sites of YB-1 for each of the identified molecules. This docking was used to avoid the selection of compounds that couldn't fit in the pocket, and to compute a first estimation of the ligand's affinity to the target.

### Docking

Docking computations were performed using AutoDock 4.2.6 (**Morris et al., 2009**), with atom types grids generated using AutoGrid 4.2.6. Grid box was 74x66 × 82 points of size, centered on the binding site, with a spacing of 0.375 Å. Each docking computations performed 100 runs of the genetic algorithm, with a maximum number of generations and energy evaluations equal to 27,000 and 2,500,000, respectively.

### Pharmacophoric screening

The pharmacophore library was built using the commercially available 'all-stock screening compounds' dataset from MolPort company. The corresponding 3D conformers were generated using Open Babel 2.3.2 (**O'Boyle et al., 2011**). Pharmacophore generation and pharmacophore 3D alignments were performed using Align-it software from Silicos-it (**Taminau et al., 2008**). The pharmacophore representation is composed of eight different types of pharmacophoric points: lipophilic region, hydrogen bond donor, hydrogen bond acceptor, positive charge center, negative charge center, hybrid type of aromatic and lipophilic, hybrid type of hydrogen bond donor and acceptor.

A total of 208 million pharmacophores were screened, representing around 7.3 million distinct molecules. The pharmacophore alignments are scored based on point types, and the overlap volumes. As a result, four metrics were computed:

1. PP: the number of pharmacophoric points that are in common between the reference representation and the database representation;
2. REF: the « percentage » of the volume of the pharmacophores generated from the reference molecule that is common and aligned to the pharmacophores from our database (ranging from 0 to 1);
3. DB: is the « percentage » of the volume of the pharmacophores generated from our database that is common and aligned to the pharmacophores from the reference molecule (ranging from 0 to 1);
4. Tanimoto: represents the similarity between the two pharmacophores (ranging from 0 to 1).

At the end, the top molecules based on Tanimoto metrics were selected for each screening, with a minimum threshold for PP.

## ADME-T prediction

ADME-T predictions were performed using SAR/QSAR models from ADMETlab (*Dong et al., 2018*). In these models, the prediction is based on molecular descriptors computed from SMILEs.

## II. Virtual Screening of FDA-Approved Drugs using MTiOpenScreen

For the drug repurposing part of our study, an automated blind docking of an FDA-approved drug library (Drugs-lib) (*Lagarde et al., 2018*) was considered using MTIiOpenScreen (*Labbé et al., 2015*), a web server that performs virtual screening using AutoDock Vina (*Trott and Olson, 2010*). The Drugs-lib library contains 7173 stereoisomers corresponding to 4574 single isomer drugs. A gradient-based conformational search approach is used and defines the search space by a grid box that was centered at the center of our protein and its dimensions were 20 Å in x, y, and z. The grid resolution is internally assigned to 1 Å. A number of binding modes of 10 and an exhaustiveness of 8 were used. The scoring of the generated docking poses and ranking of the ligands is based on the Vina empirical scoring function approximating the binding affinity in kcal.mol−1.

## III. Physico-Chemical and Purchasability Filters

We applied physico-chemical filters to select molecules belonging to a preferred chemical space that has drug-like properties. This included compounds in the ranges: $250 < MW$ (Molecular Weight) $< 650$; $0 < tPSA$ (topological polar surface area) $< 180$; $-3 < logP < 6$; $0 < $ number of HBD (hydrogen bond donors) $< 7$; $0 < $ number of HBA (hydrogen bond acceptors) $< 12$; $0 < $ Rotatable Bonds $< 10$; $0 < $ Rigid Bonds $< 30$; Num Rings $\leq 6$; Max Size Ring $\leq 18$; $3 < $ Num Carbon Atoms $< 35$; $1 < $ Num HeteroAtoms $< 15$; $0.1 < $ Ratio H/C $< 1.1$; Num Charges $\leq 3$; $-2 < $ Total Charge $< 2$. Compounds F5 and F6 were an exception for some of these criteria. We also made sure that the selected compounds were commercially available with a purity >95%.

## Appendix 3

### MT bench: Image acquisitions and statistical analysis

Images of the cellular fluorescent signals were acquired on the high content imaging system Opera Phenix Plus from Perkin Elmer on 40 x water immersion objectives with a numerical aperture of 1.1, allowing us to obtain a good resolution in the confocal mode. 160 fields of views were taken for each well resulting in thousands of cells to be analyzed, by well, in a 96-well plate format in order to have the strongest statistical significance. The data were calculated and extracted with the HARMONY software version 5.0 using an analysis pipeline containing successive building blocks for image segmentation, selection of population of interest, and calculation of signal enrichment on the microtubules (MTs) (*Figure 3—figure supplement 2.3*). The enrichment is calculated on identified spots using the GFP channel that corresponds to the signal of the bait protein forced to be localized at MT due to its fusion with a microtubule binding domain (MBD). Spots representing segments of MTs were selected based on their shape and on the intensity of the GFP channel signals (corresponding to the presence of the bait on MT). mRNA were detected with Cy3-labelled poly(dT) and could be brought on MT due to their potential interaction with the bait. The calculated GFP and Cy3 intensities in the spots and in the cytoplasm were extracted from the HARMONY software and treated subsequently in order to measure the slope of the mRNA enrichment on MTs (mean spot intensity divided by mean cytoplasm intensity versus mean bait spot intensity) (*Figure 3c*). The robustness of a screening assay is usually determined according to the value of a calculated SSMD (Strictly standardized mean difference). The SSMD measures the strength of the difference between two controls following *Equation 6*:

$$SSMD = \frac{(\mu_n - \mu_p)}{\sqrt{(\sigma_n^2 + \sigma_p^2)}} \tag{6}$$

where $\mu_p$ and $\sigma_p^2$ are the mean and standard deviation values of the positive control and $\mu_n$ and $\sigma_n^2$ are those of the negative control. If the difference between the mean values is many times greater than the standard deviation, the assay is accurate. An assay with an SSMD value ≥7 is considered of excellent quality and with an extremely strong control.

## Appendix 4

## NMR data analysis

### I.YB-1:Ligand binding (complex formation)

Data analysis was performed on the 15 ligands and 20 residues exhibiting significant CSPs. The ligands and residues in question are shown in *Figure 5*. For this, two data mining approaches were used: principal component analysis (PCA) and a correlation-matrix-based hierarchical clustering. First, the chemical shift data from each spectrum were represented as a one-dimensional vector that contains $\Delta\delta_{avg}$, that corresponds to the normalized δN and δH values. Following this, 15 vectors from ligands and 20 residues were concatenated to build a two-dimensional matrix. Some row vectors lacking standardized chemical shift data due to disappearing NMR signals were replaced by a high CSP value of 0.1 in order to mark a different exchange regime. The matrix size was 300 [20 residues ×15 ligands]. In order to look at the changes affecting the residues as a function of the ligand and vice-versa, the above analysis approaches were performed on this matrix, denoted $A$, and on it transpose, denoted $A^T$ [15 ligands ×20 residues].

A PCA standard singular value decomposition analysis was performed on matrix A and on the transpose $A^T$. PCA is a statistical method widely used in exploratory data analysis (*Pearson, 1901*). This non-parametric method compresses the dimension of a matrix by finding the directions that captures most of the variability in our data matrix and thus can reveal some simplified structures hidden in the dataset. For matrix $A$, results show that the first 6 PC dimensions represent 97% of the variance, meaning that these 6 PCs are likely to describe most contributions of the signal changes. And for the transpose $A^T$, results show that the first 5 PCs represent 95% of the variance. The results are illustrated in *Figure 5*. Outliers were detected using SPE and Hoteling's $T^2$ tests. These two tests are complementary to each other. A clustering analysis of the PC results was also conducted using $k$-means.

The matrices were also analyzed using a correlation-matrix-based hierarchical clustering, where Pearson's correlation coefficient was calculated based on the ligand-induced CSPs followed by an agglomerative hierarchical clustering to extract multiple correlation patterns. Pearson's correlation is used to measure similarity between different rows/columns. And the cluster analysis seeks to build a hierarchy of clusters, where each observation starts in its own cluster, and pairs of clusters are merged as one moves up the hierarchy. Features are thus grouped hierarchically according to their distances. Threshold was set at 0.7 and we were able to see five different clusters grouped in the main diagonal for matrix $A$ and four for $A^T$. Thus, the corresponding data in *Figure 5— figure supplement 1* are represented as correlation heatmaps arranged using a dissimilarity matrix, which gives information on how far are two features, to improve the visual representation, and the relationship between features is illustrated in a dendrogram. For negative and positive correlations, the distance will be close to zero. If there is no correlation, the distance will be 0. The results obtained with the correlation-matrix-based hierarchical clustering are in line with PCA analysis. P1, C2 and F2 (black) are again found as outliers with different effect on the binding. Flavonols F1, F3 and F7 show an identical behavior. And the rest of the molecules represent a more diverse but similar behavior. When looking at $A^T$, residues W65, V84, F85, V86, G119, A120 and E121(green) form a highly correlated cluster. The first outliers with 100% similarity are D83 and E117 (red), followed by K118 and G116 (blue). F74, H87 and W65sc (grey) also manifest being neighbors. Even though we were able to identify the interacting pocket, and classify the ligands by binding mode, compared to the PCA analysis, this technique is less sensitive and less informative in extracting specific residue information related to ligand selectivity.

### II. Ligand-RNA Competitive Binding to YB-1

The Ligand's ability to compete with RNA on YB1 binding was evaluated by comparing pair displacement vectors of affected residues using their scalar product (denoted $SP(residue)$). These vectors correspond to the chemical displacement induced after adding the ligand ($\vec{u}$), RNA ($\vec{v}$) and both RNA +ligand ($\vec{w}$) to YB-1 (see *Figure 7a*, example for P1). The aim is to compare the following pair ($\vec{u}$-$\vec{w}$-$\vec{w}$) in order to see if the induced chemical shift displacement of YB-1 residues is closer to the YB-1 Ligand-bound state or to the YB-1 RNA-bound state.

And thus,

$$SP(residue) = f(\vec{u}, \vec{v}, \vec{w}) = (\vec{u} - \vec{w}).(\vec{v} - \vec{w}) \tag{7}$$

1. If $SP(residue) > 0$, this means that the angle formed by these two pair vectors is acute and that vectors $\vec{w}$, $\vec{v}$ and $\vec{u}$ move in the same direction, which can be translated in to the fact that the concerned residue show an additive effect between the ligand and the RNA.

2. If $SP(residue) < 0$, this means that the angle formed by these two pair vectors is obtuse and that vectors $\vec{w}$, $\vec{v}$ and $\vec{u}$ move in different directions and that vector $\vec{w}$ is closer to $\vec{u}$ rather than $\vec{v}$, which can be translated into the fact that the concerned residue shows a significant competition of the ligand on the binding site rather than an additivity.

3. If $SP(residue) = 0$, can mean three things:

Either that $(\vec{u} - \vec{w}) = 0$; $\implies \vec{u} = \vec{w}$; meaning that the concerned residue represents the exact displacement for YB-1 +Ligand and for YB-1 +RNA + Ligand. Here, we have a full competition where the Ligand's effect manifest at 100%. This is the case of residue A120.

Or $(\vec{v} - \vec{w}) = 0$; $\implies \vec{v} = \vec{w}$; meaning that the concerned residue represents the exact displacement for YB-1 +RNA and for YB-1 +RNA + Ligand. Here, the ligand does not compete with RNA. No residue representing this case was observed.

Or $(\vec{u} - \vec{w})$ is perpendicular to $(\vec{v} - \vec{w})$; meaning that the concerned residue reflects the ligand's competitive binding. This is the case of residue V114.

This is an unsupervised and systematic way to compare two displacements for each residue. We could on top of this reduction of dimensionality apply a clustering algorithm to identify different clusters. However, this is beyond the scope of this application since we have results for only two ligands (P1 and C8).

## Appendix 5

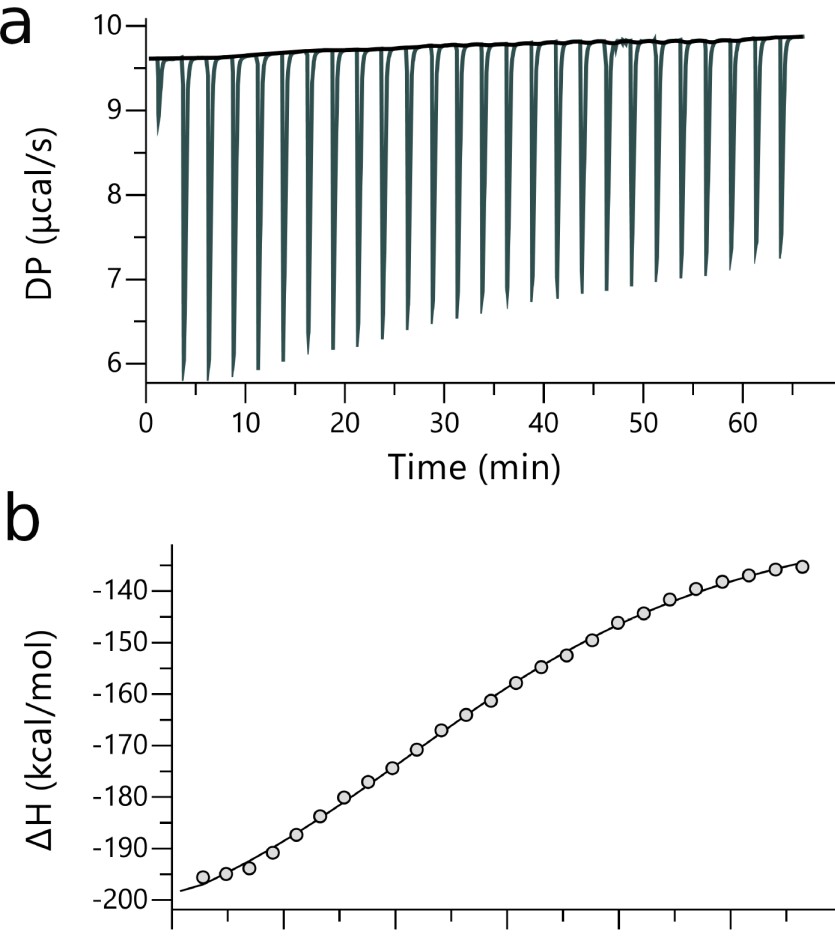

**Appendix 5—figure 1.** Isothermal Titration Calorimetry measurements of YB-1/P1 binding. (**a**) ITC raw data for titration of P1 into YB-1 in aqueous buffer solution. (**b**) Enthalpograms retrieved from (**a**). The solid line represents the fit to a single-site binding model.

**Appendix 5—table 1.** ITC measurements of YB1/P1 complex.
Calorigrams of raw data are provided in *Appendix 5—figure 1*.

| | [Ligand] (M) | [YB-1] (M) | N | Kd (M) | ΔH (kcal. mol— 1) | ΔG (kcal. mol— 1) | -TΔS (K.kcal. mol— 1) | Red.$\chi^2$ |
|---|---|---|---|---|---|---|---|---|
| P1 | $200.10^{-6}$ | $14.10^{-6}$ | $1.54 \pm 3.9 \times 10^{-2}$ | $5.84 \times 10^{-9} \pm 674 \times 10^{-9}$ | $-98.1 \pm 4.61$ | $-7.14$ | $91.00$ | $0.47$ |

The online version of this article includes the following source data for appendix 5—table 1:

• **Appendix 5—table 1—source data 1.** ITC raw and fitted data obtained from P1 binding to YB-1.

• (see legends of *Appendix 5—figure 1* and *Appendix 5—table 1*).

**Appendix 5—table 2.** Thermodynamics of P1 binding to YB-1 obtained from ABFE simulations and ITC experiments.
Comparison of P1/YB-1 binding free energies (ΔG) and their respective enthalpic (ΔH) and entropic (ΔS) contributions from ITC and computations. ΔS was calculated according to ΔG = ΔH - TΔS at 298 K; units are in kcal.mol−1.

| | $\Delta G_{bind}$ (kcal.mol— 1) | $\Delta H_{bind}$ (kcal.mol— 1) | $-T\Delta S_{bind}$ (K.kcal.mol— 1) |
|---|---|---|---|
| ITC | −7.14± 0.47 | −98.10± 4.61 | 91.00 |
| ABFE | −7.24± 0.52 | −28.62± 1.62 | 21.38 |

**Appendix 5—table 3.** Supplier list and compound IDs of the 40 tested molecules.

| Compound | Traditional name | Compound ID | Supplier |
|---|---|---|---|
| F1 | Quercetin | MolPort-001-740-557 | MolPort |
| F2 | Quercetagetin | MolPort-006-147-776 | MolPort |
| F3 | 3-O-methylquercetin | 90081 | SigmaAldrich |
| F4 | Fisetin | MolPort-000-882-130 | MolPort |
| F5 | Rutin | MolPort-001-740-246 | MolPort |
| F6 | myricitrin | 91255 | SigmaAldrich |
| F7 | Herbacetin | MolPort-019-998-217 | MolPort |
| F8 | Vincetoxicoside B | MolPort-035-758-036 | MolPort |
| F9 | Scutellarien | MolPort-003-724-680 | MolPort |
| F10 | Luteolin-7-methylether | MolPort-001-740-950 | MolPort |
| F11 | Naringenin | MolPort-000-861-091 | MolPort |
| C1 | Butein | MolPort-006-111-425 | MolPort |
| C2 | Okanin | FO66168 | CarboSynth |
| C3 | Robtein | Amb22172818 | Ambinter |
| C4 | Chorilifol B | MolPort-039-338-845 | MolPort |
| C5 | Bavachalcone | FB145210 | CarboSynth |
| C6 | homobutein | FM65711 | CarboSynth |
| C7 | Cardamonin | FC66017 | CarboSynth |
| C8 | 3',3,4,5'-tetrahydroxychalcone | FC66017 | CarboSynth |
| C9 | 2',4'-dihydroxy-4-methoxychalcone | MolPort-000-662-842 | MolPort |
| C10 | 2'-hydroxy-4'-methoxychalcone | MolPort-000-779-850 | MolPort |
| C11 | 2',4,4'-trihydroxychalcone | MolPort-001-741-660 | MolPort |
| C12 | Lichochalcone B | MolPort-046-594-311 | MolPort |
| A1 | 10–3-[2-(2,3-dihydro-1-benzofuran-5-yl)ethoxy]phenyl-2-(3,4-dihydroxyphenyl)–3,5-dihydroxy-9H,10H-pyrano[2,3 h]chromene-4,8-dione | MolPort-035-700-332 | MolPort |
| A2 | 2-(3,4-dihydroxyphenyl)–3,5-dihydroxy-10-(naphthalen-1-yl)–9 H,10H-pyrano[2,3 h]chromene-4,8-dione | MolPort-029-885-579 | MolPort |
| A3 | [8-(4-chlorophenyl)–2-(3,4-dihydroxyphenyl)–3,5-dihydroxy-4-oxofuro[2,3 h]chromen-9-yl]acetic acid | MolPort-044-544-604 | MolPort |
| A4 | 2-(3,4-dihydroxyphenyl)–3,5-dihydroxy-10–2-[2-(4-methoxyphenyl)ethoxy]phenyl-9H,10H-pyrano[2,3 h]chromene-4,8-dione | MolPort-035-699-845 | MolPort |
| A5 | 2-(3,4-dihydroxyphenyl)–3,5-dihydroxy-10-isopropyl-9H,10H-pyrano[2,3 h]chromene-4,8-dione | MolPort-029-886-488 | MolPort |
| A6 | 6,8-dibromo-2-(3-chloro-4-hydroxy-5-methoxyphenyl)chromen-4-one | MolPort-008-821-914 | MolPort |

*Appendix 5—table 3 Continued on next page*

*Appendix 5—table 3 Continued*

| Compound | Traditional name | Compound ID | Supplier |
|---|---|---|---|
| A7 | 6,8-dibromo-3-hydroxy-2-(4-hydroxy-3-methoxyphenyl)chromen-4-one | MolPort-023-282-651 | MolPort |
| A8 | 2-(4-hydroxy-3-iodo-5-methoxyphenyl)chromen-4-one | MolPort-002-521-806 | MolPort |
| P1 | Niraparib | MolPort-023-219-142 | MolPort |
| P2 | Olaparib | MolPort-009-679-395 | MolPort |
| P3 | Talazoparib | MolPort-028-600-028 | MolPort |
| P4 | Veliparib | MolPort-016-633-168 | MolPort |
| P5 | Rucaparib | MolPort-028-744-762 | MolPort |
| D1 | Nebivolol | MolPort-015-163-751 | MolPort |
| D2 | Mefloquine | MolPort-006-170-692 | MolPort |
| D3 | Icotinib | MolPort-039-139-676 | MolPort |
| D4 | Cabotegravir | MolPort-035-944-338 | MolPort |

**Appendix 5—table 4.** Primers used for RT-PCR analysis.
Sequences are from 5′ to 3′.

| | Forward | Backward |
|---|---|---|
| canx | GCAACCACTTCCCTTCCAT | TCCGCCTCTCTCTTTACTGC |
| calr | TGTCAAAGATGGTGCCAGAC | ACAACCCCGAGTATTCTCCC |
| oaz1 | TACAGCAGTGGAGGGAGACC | GGATAAACCCAGCGCCAC |
| rpl8 | AGATGGGTTTGTCAATTCGG | CAAGAAGACCCGTGTGAAGC |
| eif4g1 | CCCAACTGTAGAAGGCATCC | CTCCAGGCCCTTGTAGTGAC |
| fnbp1 | GCATGAAGTTATCTCCGAGAACA | CGGCCATCGTGAAAGTTTGAT |
| nin | GGAGGAACTCACCGACCTTTG | CGTCCGTAACGCTTCCCAC |
| cdkal1 | GGGACTGAGTATCATTGGGGT | CCAAGCCGCCTTCCATTATC |
| mkln1 | AGCCACGATGGAGTCAAATCA | TGGCACTAGGACCATTCTCTTT |
| eif4g2 | AATCGCACTCTCCACTTTGG | GCTGCTGAGTTCTCGGTGA |
| ubl3 | TGACAATTGGCCAATGGACTG | GCCACCAAATGCATCACTGT |
| actin | CATGTACGTTGCTATCCAGGC | CTCCTTAATGTCACGCACGAT |
| gapdh | CCTCCTGCACCACCAACTGCTTA | GTGATGGCATGGACTGTGGTCAT |

**Appendix 5—table 5.** RT-PCR analysis of 13 mRNAs isolated via MT bench pull-down or via magnetic beads RIP and for 3 different RBPs (YB-1, HuR, and FUS).

| | MT bench | | | IP (beads) | | |
|---|---|---|---|---|---|---|
| mRNA\ Bait | YB-1 | HuR | FUS | YB-1 | HuR | FUS |
| fnbp1 | 0.733 | –0.339 | 1.032 | 0.429 | 0.227 | 0.641 |
| rpl8 | –0.436 | –1.647 | –1.976 | 0.098 | –2.546 | –1.232 |
| eif4g1 | 1.697 | 0.009 | 0.024 | 1.527 | 0.176 | 1.258 |
| gapdh | –0.228 | –2.772 | –2.834 | 0.294 | –3.616 | –1.609 |
| ubl3 | 0.356 | 1.439 | 2.073 | 0.094 | 1.297 | 1.658 |
| canx | –0.227 | 1.013 | 2.286 | –0.367 | 1.555 | 0.770 |

*Appendix 5—table 5 Continued*

| | MT bench | | | IP (beads) | | |
|---|---|---|---|---|---|---|
| actin | −1.661 | −0.855 | −3.051 | −2.007 | −0.074 | −1.716 |
| eif4g2 | 0.546 | −0.118 | 1.458 | 1.117 | 0.037 | 0.324 |
| nin | 0.906 | 0.371 | 2.806 | 0.859 | 0.824 | 2.259 |
| calr | 0.403 | −0.405 | −1.801 | −0.357 | 0.893 | −1.163 |
| cdk1 | 0.137 | 1.050 | 0.569 | 0.954 | 1.558 | 0.896 |
| oaz1 | −1.783 | −0.940 | −3.758 | −1.310 | −0.976 | −2.189 |
| mkln1 | 0.418 | 0.196 | 1.974 | −0.127 | −0.256 | 0.103 |

