## [Editor Report]

A novel approach is introduced to modulate and/or inhibit protein-RNA interactions, based upon integration of computational techniques with cellular assays.

---

## [Decision Letter]

**Decision letter after peer review:**

Thank you for submitting your article "Targeting RNA:Protein Interactions with an Integrative Approach Leads to the Identification of Potent YB-1 Inhibitors" for consideration by *eLife*. Your article has been reviewed by 3 peer reviewers, including Nir Ben-Tal as Reviewing Editor and Reviewer #1, and the evaluation has been overseen by and José Faraldo-Gómez as Senior Editor.

Essential revisions:

1. We are concerned about the specificity of the MT bench assay for evaluating RPIs. For example, if a non-RBP were recruited to the microtubule would the authors detect RPIs? If not, what is the difference in signals between authentic RBP-RNA RPIs compared with non-specific interactions of proteins using the MT bench assay? We only see RBPs used as proof of concept for the RPI assays in the current manuscript. We think the inclusion of negative controls (non-RBP proteins) is critical for the application of this screen to evaluate YB-1 ligands. How specific is this methodology for assaying RNA-compared with DNA-binding proteins? This could be another control experiment that the authors consider to further strengthen the manuscript.

2. While the workflow leading to the demonstration and validation of the YB-1 RNA-protein interaction inhibitor (the FDA-approved PARP inhibitor) is well developed and strongly supported by data, what is lacking is evidence that the observed effect is specific to inhibition of YB-1-mediated regulation of translation and whether the expression of transcripts specifically regulated by YB-1 is affected. YB-1 may be regulating the translation of a large number of transcripts, but it is not a general translation factor such as an eIF or eEF. Therefore showing a general reduction in cellular protein synthesis, by puromycin incorporation assays, might not be appropriate and may rather be misleading. Instead, the protein expression of a number of known mRNA targets of YB-1 should be checked by western blotting, in the absence and presence of the inhibitor(s), to demonstrate that the translation inhibitory effect is actually there and is specific to YB-1. This can be further reinforced by reporter gene assays, using reporter gene constructs containing 3'UTRs of transcripts which are known targets of YB-1. These experiments should be done also to establish the dose-dependence of the inhibitor.

3. The effect of the inhibitor on cellular function needs to be studied in greater detail to judge both the specific phenotypic effect as well as the non-specific cytotoxic effect of the inhibitor. Cell proliferation assays and apoptosis assays should be done to check the effect on cell proliferation and death. And whether overexpression of YB-1 in the presence of the inhibitor can rescue these effects should also be tested.

4. The mode of data presentation in the paper is sometimes confusing and does not allow the real evaluation of the evidence. Particularly, the representation of correlation data of mRNA coprecipitation by microtubules and RIP, for example in Figure 3D, is non-informative. Rather a much simpler and more explicit form of bar diagrams, showing fold enrichment of the mRNAs compared to controls should be represented, which will allow a much better comparison of the efficacy of the two methods. Overall, the data presentation can be much simplified and the figures seem overburdened with information.

5. The manuscript presents a blend of computation and experimental work to tackle a challenging class of proteins. As the authors pointed out in the introduction, methodologies for screening RBPs are largely lacking, and approaches complementary to the existing state of the art are needed. However, we are not convinced that the presented platform is ready to be considered a 'general' RBP screening method. The major criticism is that all the studies in the current manuscript were focused largely on a single RBP. YB-1 was also already reported in a prior manuscript from the same group (Boca et al. 2015). The validation of small molecules against RBPs is challenging and it makes sense to use tractable RBP targets for proof of concept. However, the authors are making general claims for this platform to screen for RBP inhibitors, which we do not believe they have sufficient supporting data. The authors should demonstrate the identification of ligands with reasonable specificity against additional RBP targets. The RBP family is quite large (hundreds to thousands of members) and many existing databases are available from RNA interactome capture studies that should guide the authors to select candidates. Alternatively, authors should tone down statements about the generality of the method.

[Editors’ note: further revisions were suggested prior to acceptance, as described below.]

Thank you for resubmitting your work entitled "Targeting RNA:Protein Interactions with an Integrative Approach Leads to the Identification of Potent YBX1 Inhibitors" for further consideration by *eLife*. Your revised article has been evaluated by two of the original reviewers. The reviewers conclude that manuscript has been improved; however, there are some remaining issues that need to be addressed, as outlined below.

*Reviewer #2 (Recommendations for the authors):*

The revised version of the manuscript "Targeting RNA:Protein Interactions with an Integrative Approach Leads to the Identification of Potent YBX1 Inhibitors" addresses some of the reviewers' concerns, especially about the specificity of the RNA-protein interactions detected by the MT bench assay by using DNA-binding proteins to demonstrate that the interaction is specific to RNA-binding protein YB1. However, the data regarding the cellular and functional effects of the inhibition of YB1-RNA interaction by the PARP inhibitor P1 is weak and inconclusive. Especially the data shown in Figure 8- figure supplement 3 is not at all clear to me. The figure purportedly shows "P1 but not P2 reduces Vimentin expression level in YB1-rich HeLa cells". Firstly, I do not see a downregulation of vimentin expression upon siRNA-mediated knockdown of YB1. There is an IF image with siRNA YB1 but not with control siRNA, so the extent of YB1 knockdown cannot be estimated. Secondly, even accepting that the red fluorescence is from vimentin, there is no indication in the image that it goes down upon knockdown of YB1. There is no image or western blotting of vimentin in presence and absence of P1 treatment. Western blotting is essential to determine whether a specific protein is being downregulated as an IF signal can be generated non-specifically, especially without an isotypic antibody control. Finally P1 does not seem to have a significant effect on E-cadherin expression compared to P2. The authors have therefore just tested two proteins which are regulated by YB1, and they have a 50% success rate in showing the effect of P1 on YB1-mediated regulation of protein expression, which I do not think can be called significant.

The cell proliferation study results, in Figure 8a, are also not clear. Firstly, only one time point has been studied, which does not make it a real cell proliferation assay. The change in cell number can be due to cell proliferation or cell death or a combination of both. All three compounds P1, P2 and P3 are PARP inhibitors and therefore reduce cell number. However, I do not understand the rationale why P1, which is supposed to be an YB1 inhibitor as per this manuscript, should increase the cell number in YB1 siRNA transfected cells compared to control siRNA transfected cells. If YB1 is already depleted, further inhibition of YB1 activity should further accentuate the effect of reduction in cell number, not reverse it. The rationale of showing the effect of inhibition on an already depleted molecule is not clear. It appears to me the only logical way to show that the effect is mediated by the protein is to do a rescue experiment by overexpressing the protein in the background of inhibition and show that the effect of the inhibition is reversed.

The comparison between the RNA pulldown by MT bench assay and RIP, shown in Figure 3D shows that out of 12 mRNA checked, 6 do not show correlation or show anti-correlation between the MT assay and RIP, both in case of HuR and YB1. Therefore, the data shown in this figure, which was not really apparent in the Figure 3-figure supplement 4 do not really support the authors' contention that the MT bench system and RIP gives comparable results. I do not even think that we should expect similar results, considering that the assays are very different and one happens in cells and the other in cell lysates, and it may be advisable not even to claim the same.

Overall it appears that the data for the inhibition of RNA-protein interaction between YB1 and cellular RNAs by P1 is strong, and is an important validation of the methodology. On the other hand the data about the functional effect of inhibition of YB1-RNA interaction is inconclusive or absent. P1 being a PARP inhibitor, it is particularly important to show that the effect being seen is due to specific inhibition of YB-1-mediated regulation of gene expression. In the absence of such evidence I think it would be better to tone down, or completely remove, the claims about the functional effects of YB1 inhibition, and just focus on the inhibition of YB1 RNA-protein interactions in the manuscript.

---

## [Author Response]

Essential revisions:1. We are concerned about the specificity of the MT bench assay for evaluating RPIs. For example, if a non-RBP were recruited to the microtubule would the authors detect RPIs? If not, what is the difference in signals between authentic RBP-RNA RPIs compared with non-specific interactions of proteins using the MT bench assay? We only see RBPs used as proof of concept for the RPI assays in the current manuscript. We think the inclusion of negative controls (non-RBP proteins) is critical for the application of this screen to evaluate YB-1 ligands. How specific is this methodology for assaying RNA-compared with DNA-binding proteins? This could be another control experiment that the authors consider to further strengthen the manuscript.

We agree with the reviewers that additional negative control experiments will further strengthen the manuscript. Indeed, in the previous version of the manuscript we only used GFP, a non-RBP, as a negative control in our assays and showed that it did not bring mRNA to microtubules and no RPIs were detected. To provide additional negative controls, we have followed the reviewers interesting suggestions and used DNA-binding proteins as baits to analyze whether mRNA is brought nonspecifically onto microtubules in the MT bench assay. These proteins are TOP1, APE1, and LIG1. Our results indicate that DNA-binding proteins indeed fail to bring mRNA onto the microtubule in the MT bench assay. In contrast, we observed that YB-1 successfully brought mRNAs onto microtubules. This additional analysis is now included in the revised manuscript on page 8 and illustrated in a new Figure 3—figure supplement 3.

2. While the workflow leading to the demonstration and validation of the YB-1 RNA-protein interaction inhibitor (the FDA-approved PARP inhibitor) is well developed and strongly supported by data, what is lacking is evidence that the observed effect is specific to inhibition of YB-1-mediated regulation of translation and whether the expression of transcripts specifically regulated by YB-1 is affected. YB-1 may be regulating the translation of a large number of transcripts, but it is not a general translation factor such as an eIF or eEF. Therefore showing a general reduction in cellular protein synthesis, by puromycin incorporation assays, might not be appropriate and may rather be misleading. Instead, the protein expression of a number of known mRNA targets of YB-1 should be checked by western blotting, in the absence and presence of the inhibitor(s), to demonstrate that the translation inhibitory effect is actually there and is specific to YB-1. This can be further reinforced by reporter gene assays, using reporter gene constructs containing 3'UTRs of transcripts which are known targets of YB-1. These experiments should be done also to establish the dose-dependence of the inhibitor.

Yes, we understand this point. As indicated in the manuscript, it is very difficult to find functional cellular assays that would reveal a phenotype specific to a general RBP such as YB-1. This is even more difficult with YB-1 since it binds nonspecifically to most mRNAs as shown from CLIP analysis^1^ (for example, in contrast with TDP-43 that binds specifically to GU-rich repeats). This was one of the reasons to develop a specific cellular assay such as the MT bench assay. YB-1 originates from cold shock proteins in bacteria which preserve global mRNA translation during cold stress, presumably by removing secondary structures. YB-1 in contrast with many RBPs has only a single structured RNA-binding domain, which is not favorable for a specific binding to some mRNA sequences/structures. As noticed by the reviewers, YB-1 is indeed not a general translation factor but is a general protein that binds to most non polysomal mRNA ^2^. mRNAs, even those highly translated, switch from a polysomal state (active) to a non polysomal state (dormant) from time to time. In a recent work, we showed that YB-1 prepares non polysomal mRNAs in such a way to facilitate the translation from dormant to active state. We also showed that, accordingly, decreasing the expression of YB-1 reduces global mRNA translation rates in HeLa cells^3^. Consistent with this trend, a global decrease of mRNA translation as observed with Niraparib (P1) that targets YB-1 makes sense. We have no knowledge of established 3’UTRs which would be highly specific to YB-1. YB-1 binds non specifically to both mRNA coding sequences and 3’UTRs (YBX1 data^1^, YBX3 data^4^).

Large scale and in depth analysis should be performed to find out whether specific structures/sequences increase significantly the YB-1 dependency in mRNA translations. However, as noticed by the reviewers, the expression of some proteins associated to malignancy have been associated to YB-1 expression level notably Vimentin and E-cadehrin^3^. For this we performed a new experiment where we measured the expression levels of these two proteins after silencing YB-1 expression in HeLa cells, in the absence and in the presence of Niraparib P1 and Olaparib P2 (P2 is used here as a negative control). Results show that Niraparib, but not Olaparib, decreases the dependence on YB-1 of Vimentin expression level (significant) and that of E-cadherin (non-significant). Other proteins such as eIF5a and RPL36, used here as negative controls, did not show a similar behavior. These results were thus in agreement with a specific effect of Niraparib on YB-1-controlled protein expression, and with a recent report^5^ showing the down regulation of Vimentin expression in ovarian cancer cells when treated with Niraparib.

This is now discussed on page 18 of the revised manuscript and the new data are included as a new Figure 8—figure supplement 3.

1. Wu, S.-L. et al. Genome-wide analysis of YB-1-RNA interactions reveals a novel role of YB-1 in miRNA processing in glioblastoma multiforme. Nucleic acids research 43, 8516-8528 (2015).

2. Singh, G., Pratt, G., Yeo, G.W. and Moore, M.J. The clothes make the mRNA: past and present trends in mRNP fashion. Annual review of biochemistry 84, 325 (2015).

3. Budkina, K. et al. YB-1 unwinds mRNA secondary structures in vitro and negatively regulates stress granule assembly in HeLa cells. Nucleic acids research 49, 10061-10081 (2021).

4. Van Nostrand, E.L. et al. A large-scale binding and functional map of human RNA-binding proteins. Nature 583, 711-719 (2020).

5. Zhen Zeng, Jing Yu, Zhongqing Jiang, Ningwei Zhao, "Oleanolic Acid (OA) Targeting UNC5B Inhibits Proliferation and EMT of Ovarian Cancer Cell and Increases Chemotherapy Sensitivity of Niraparib", Journal of Oncology, vol. 2022, 12 pages, 2022. https://doi.org/10.1155/2022/5887671

3. The effect of the inhibitor on cellular function needs to be studied in greater detail to judge both the specific phenotypic effect as well as the non-specific cytotoxic effect of the inhibitor. Cell proliferation assays and apoptosis assays should be done to check the effect on cell proliferation and death. And whether overexpression of YB-1 in the presence of the inhibitor can rescue these effects should also be tested.

We agree with the reviewers on this remark. YB-1 is associated with the high proliferation rate of cancer cells (and silencing YB-1 does not induce apoptosis). Therefore, we performed cell proliferation assays using cells treated with siRNA and siNEG allowing us to manipulate the endogenous YB-1 expression level rather than a more artificial rescue experiment. These assays were performed in the presence of 3 PARP-1 inhibitors at low concentrations: Niraparib (P1) our hit, and two negative controls Olaparib (P2) and Talazoparib (P3). We used a 48 h incubation time which allows to observe effects at lower concentration of compounds. All PARP-1 inhibitors decease cell proliferation, albeit to a higher extent with P3. However, P2 or P3 further decrease cell proliferation in siRNA-treated cells compared to siNEG-treated cells (significant differences at 5 µM). In contrast, P1 rather further decreases cell proliferation in siNEG-treated cells when YB-1 levels are high (non-significant variations but opposite to those observed with P2 and P3). This new result is now presented as new Figure 8a. In addition, we show that the separation distance between cells increases significantly in YB-1-rich cells treated with P1, in contrast to P2 and P3 (significant differences) (new Figure 8—figure supplement 1). A short distance of separation between cells may be due to colony formation when cells were plated at low density and allowed to grow for 48 h. Again, it means that Niraparib better inhibits cell proliferation in YB-1-rich cells when compared with what is observed with the two other PARP inhibitors P2 and P3. The text on pages 16 and 18 was rewritten to put this in evidence.

4. The mode of data presentation in the paper is sometimes confusing and does not allow the real evaluation of the evidence. Particularly, the representation of correlation data of mRNA coprecipitation by microtubules and RIP, for example in Figure 3D, is non-informative. Rather a much simpler and more explicit form of bar diagrams, showing fold enrichment of the mRNAs compared to controls should be represented, which will allow a much better comparison of the efficacy of the two methods. Overall, the data presentation can be much simplified and the figures seem overburdened with information.

We have modified the presentation of the dataset in Figure 3D and limit it to YB-1 and HuR. However, the correlation data which is more visual was moved to Figure 3—figure supplement 4 as panel c. We have tried to do our best to present the figures differently and make them less crowded. For example, Figure 7 was divided in 2 separate figures: Figure 7 (with 2 panels a and b) and Figure 8 with 3 panels (a, b and c). New figures were also added in response to the reviewer’s comments and several were reorganized (for example, Figure 7c was moved to Figure 8—figure supplement 4a).

In total 5 new figures were included: Figure 3d, Figure 3—figure supplement 3, Figure 8a, Figure 8—figure supplement 1, Figure 8—figure supplement 3, and Appendix 1-Figure 1.

Reorganized figures: Figures 3, 4, 7 and 8, Figure 3—figure supplement 4, and Figure 8—figure supplement 4 and 5.

5. The manuscript presents a blend of computation and experimental work to tackle a challenging class of proteins. As the authors pointed out in the introduction, methodologies for screening RBPs are largely lacking, and approaches complementary to the existing state of the art are needed. However, we are not convinced that the presented platform is ready to be considered a 'general' RBP screening method. The major criticism is that all the studies in the current manuscript were focused largely on a single RBP. YB-1 was also already reported in a prior manuscript from the same group (Boca et al. 2015). The validation of small molecules against RBPs is challenging and it makes sense to use tractable RBP targets for proof of concept. However, the authors are making general claims for this platform to screen for RBP inhibitors, which we do not believe they have sufficient supporting data. The authors should demonstrate the identification of ligands with reasonable specificity against additional RBP targets. The RBP family is quite large (hundreds to thousands of members) and many existing databases are available from RNA interactome capture studies that should guide the authors to select candidates. Alternatively, authors should tone down statements about the generality of the method.

We agree that this is not sufficient to generalize to all RBPs. And as noticed by the reviewers, performing a complete study for other RBPs would require a separate study. Therefore, we tuned down statements about the generality of the method in the Discussion section on page 21.

We did show that we can detect mRNA-RBP interactions with two other RBPs HuR and FUS (Figure 3d) and used them as a control to show the specificity of the tested small molecules towards YB-1 (Figure 4b,c). In the discussion, on page 21, we now explain that YB-1, because it has a single cold-shock domain and a druggable pocket, is an “ideal” target. We also explain that many RNPs harbors many RNA-binding domains, which may reduce the sensitivity of our method when a specific domain is targeted by small molecules because the other domains would contribute to the binding to mRNA. However, a single RNA-binding domain may be isolated and used as bait for the MT bench assay to overcome this obstacle. Developing molecules what would target a specific domain may be sufficient to modulate the biological function exerted by the full length protein.

[Editors’ note: further revisions were suggested prior to acceptance, as described below.]

Reviewer #2 (Recommendations for the authors):The revised version of the manuscript "Targeting RNA:Protein Interactions with an Integrative Approach Leads to the Identification of Potent YBX1 Inhibitors" addresses some of the reviewers' concerns, especially about the specificity of the RNA-protein interactions detected by the MT bench assay by using DNA-binding proteins to demonstrate that the interaction is specific to RNA-binding protein YB1. However, the data regarding the cellular and functional effects of the inhibition of YB1-RNA interaction by the PARP inhibitor P1 is weak and inconclusive. Especially the data shown in Figure 8- figure supplement 3 is not at all clear to me. The figure purportedly shows "P1 but not P2 reduces Vimentin expression level in YB1-rich HeLa cells". Firstly, I do not see a downregulation of vimentin expression upon siRNA-mediated knockdown of YB1. There is an IF image with siRNA YB1 but not with control siRNA, so the extent of YB1 knockdown cannot be estimated. Secondly, even accepting that the red fluorescence is from vimentin, there is no indication in the image that it goes down upon knockdown of YB1. There is no image or western blotting of vimentin in presence and absence of P1 treatment. Western blotting is essential to determine whether a specific protein is being downregulated as an IF signal can be generated non-specifically, especially without an isotypic antibody control. Finally P1 does not seem to have a significant effect on E-cadherin expression compared to P2. The authors have therefore just tested two proteins which are regulated by YB1, and they have a 50% success rate in showing the effect of P1 on YB1-mediated regulation of protein expression, which I do not think can be called significant.

We have decided, as proposed by Reviewer 2 in the last paragraph (point 4), to withdraw Figure 8- figure supplement 3 and all related information from the manuscript. Only analyzing the variations of a couple of genes is not enough to draw undisputed conclusions. Whether the interaction between P1 and YB-1 plays a critical role in gene expression deserves a larger scale study on its own.

The cell proliferation study results, in Figure 8a, are also not clear. Firstly, only one time point has been studied, which does not make it a real cell proliferation assay. The change in cell number can be due to cell proliferation or cell death or a combination of both. All three compounds P1, P2 and P3 are PARP inhibitors and therefore reduce cell number. However, I do not understand the rationale why P1, which is supposed to be an YB1 inhibitor as per this manuscript, should increase the cell number in YB1 siRNA transfected cells compared to control siRNA transfected cells. If YB1 is already depleted, further inhibition of YB1 activity should further accentuate the effect of reduction in cell number, not reverse it. The rationale of showing the effect of inhibition on an already depleted molecule is not clear. It appears to me the only logical way to show that the effect is mediated by the protein is to do a rescue experiment by overexpressing the protein in the background of inhibition and show that the effect of the inhibition is reversed.

Figure 8a are now presented as a supplementary figure. Figure 8a is interesting since it shows that P1 may have a different impact than P2 and P3 on the cell number depending on the level of YB-1. We understand the point raised by the reviewer regarding the rationale. When the level of YB-1 is already decreased, we may expect a higher sensitivity to P1, as proposed by the reviewer 2. On the other hand, there could be an explanation. When YB-1 levels are high, P1 may target a lot of YB-1 proteins leading to a gain of new function (toxic or cytostatic) that reduces the cell number. In this case, when YB-1 level are decreased with siRNA, P1 should have a limited impact on cell proliferation. This explanation is also in agreement with the decreased translation rate measured in cells treated with P1 when YB-1 levels are not depleted (Figure 8b). We have added a couple of sentences to discuss the point raised by the reviewer at page 19.

However, as explained by the reviewer, the functional assays are too preliminary and would deserve new experimental data to draw conclusions. We now present these data as preliminary results to stimulate future investigations (see Results section pages 16, 18 and 19).

The comparison between the RNA pulldown by MT bench assay and RIP, shown in Figure 3D shows that out of 12 mRNA checked, 6 do not show correlation or show anti-correlation between the MT assay and RIP, both in case of HuR and YB1. Therefore, the data shown in this figure, which was not really apparent in the Figure 3-figure supplement 4 do not really support the authors' contention that the MT bench system and RIP gives comparable results. I do not even think that we should expect similar results, considering that the assays are very different and one happens in cells and the other in cell lysates, and it may be advisable not even to claim the same.

Figure 3D and Figure 3-figure supplement 4 used the same data set. Of course, there are some anticorrelations for few genes such as CALR when HUR is used as bait. This is now commented in the text. We agree with the reviewer about this point. A comment has been added to the text. However, these results show that the correlation between the mRNA enrichment profiles is higher when the same RNA-binding protein is used as bait (MT-bench or RIP) than when a different RNA-binding protein is used to pull down mRNA. The enrichment profile is therefore dependent of the RNA-binding protein which is used as bait.

As noted by the reviewer, mRNA enrichment profiles were obtained after cell lysis. It is true that enrichment profile on microtubules in intact cells could be different. We now indicated this limitation in the text on page 8.

Overall it appears that the data for the inhibition of RNA-protein interaction between YB1 and cellular RNAs by P1 is strong, and is an important validation of the methodology. On the other hand the data about the functional effect of inhibition of YB1-RNA interaction is inconclusive or absent. P1 being a PARP inhibitor, it is particularly important to show that the effect being seen is due to specific inhibition of YB-1-mediated regulation of gene expression. In the absence of such evidence I think it would be better to tone down, or completely remove, the claims about the functional effects of YB1 inhibition, and just focus on the inhibition of YB1 RNA-protein interactions in the manuscript.

We agree with the reviewer. Figure 8- figure supplement 3 has been removed along with Figure 8- Fig supplement 2 and their corresponding data sources. Figure 8a has been sent to the supplementary figures (now Figure 8—figure supplement 5). We clearly indicate that functional consequences of the interaction between P1 and YB-1 should be investigated in a separate study. The results presented in the manuscript regarding the functional consequence of the interaction between P1 and YB-1 are therefore considered as the first results to initiate future research. The Results section has been partly rewritten and reordered. Our conclusions regarding this part have been tuned down (pages 3, 16, 18 and 19).